# MegaMath: Pushing the Limits of Open Math Corpora

**Fan Zhou**,* **Zengzhi Wang**\*,
**Nikhil Ranjan**, **Zhoujun Cheng**, **Liping Tang**, **Guowei He**, **Zhengzhong Liu**, **Eric P. Xing**
MBZUAI, Abu Dhabi, UAE
{koala99.zf, zzwang.nlp}@gmail.com

## Abstract

Mathematical reasoning is a cornerstone of human intelligence and a key benchmark for advanced capabilities in large language models (LLMs). However, the research community still lacks an open, large-scale, high-quality corpus tailored to the demands of math-centric LLM pre-training. We present MegaMath, an open dataset curated from diverse, math-focused sources through following practices: (1) ***Revisiting web data***: We re-extracted mathematical documents from Common Crawl with math-oriented HTML optimizations, `fastText`-based filtering and deduplication, all for acquiring higher-quality data on the Internet. (2) ***Recalling Math-related code data***: We identified high quality math-related code from large code training corpus, Stack-V2, further enhancing data diversity. (3) ***Exploring Synthetic data***: We synthesized QA-style text, math-related code, and interleaved text-code blocks from web data or code data. By integrating these strategies and validating their effectiveness through extensive ablations, MegaMath delivers **371B tokens** with the largest quantity and top quality among existing open math pre-training datasets.

## 1 Introduction

Mathematical reasoning is a fundamental yet challenging aspect of human intelligence—and a persistent difficulty for language models. Recent breakthroughs like o1 (OpenAI, 2024) and DeepSeek-R1 (Guo et al., 2025) demonstrate that, with sufficient pre-training and large-scale reinforcement learning, models can tackle competition-level math problems. However, the success of such models hinges on access to massive high-quality math pre-training datasets—e.g., DeepSeekMath's 120B tokens (Shao et al., 2024) and Qwen-2.5-Math's 1T tokens (Yang et al., 2024b). Yet no open-source dataset currently matches this scale and quality (see Table 9 for comparison), hindering progress on open math models.

A key obstacle lies in the limitations of current math web data pipelines. While web data forms the backbone of modern pre-training corpora (Penedo et al., 2024; Tang et al., 2024), existing math-specific pipelines often suffer from overly aggressive pre-filtering (*e.g.*, filtering based on HTML math tags (Paster et al., 2024)), which causes many math-relevant documents to be missed. Moreover, widely used general-purpose text extraction tools are not optimized for mathematical content—they often strip or discard equations and symbols, severely degrading data quality (Han et al., 2024; Lozhkov et al., 2024a). As a result, web-collected math data often lacks both scale and fidelity. Beyond web data, math-related code corpora (*e.g.*, AlgebraicStack (Azerbayev et al., 2023), MathCode-Pile (Lu et al., 2024)) and synthetic datasets (*e.g.*, WebInstruct (Yue et al., 2024)) have shown promising potential, but remain either limited in scale or not fully open-sourced.

To bridge this gap, we introduce MegaMath — the largest open-source English math corpus to date, totaling 371.6B tokens. It comprises 279B tokens of web data, 28.1B of code, and 64.5B of synthetic data. During its construction, we conducted extensive ablation studies and optimizations across all domains to ensure both scalability and quality. For the web domain, we designed a two-stage, coarse-to-fine extraction and filtering pipeline, improving

---

*Equal contribution.

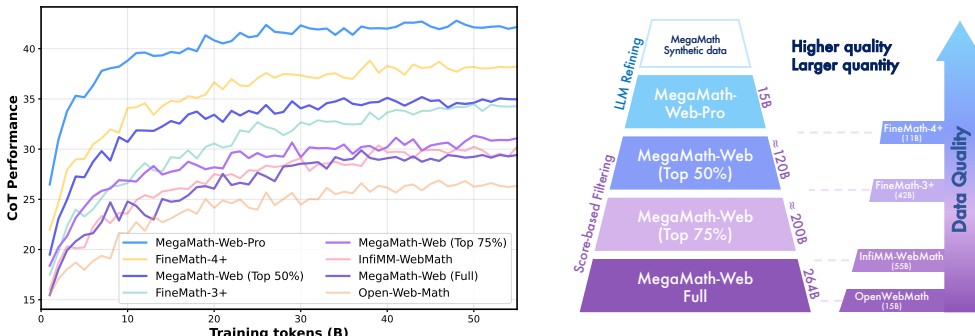

Figure 1: Comparison with existing open math corpora and MegaMath-Web subsets.

on the common pipeline. We reformatted math elements in HTML into compatible text representations (*i.e.*, LaTeX) to preserve equations and symbols during extraction. In the first stage, we applied a fast text extractor alongside a `fastText` classifier to filter candidate math documents. After deduplication, we reprocessed the retained HTMLs using a slower, high-quality extractor, followed by a second-stage `fastText` trained on the seed data from the first stage to mitigate distributional shift. This pipeline achieves both scale and fidelity, resulting in MegaMath-Web. Based on this foundation, we further developed MegaMath-Web-Pro, a premium subset delivering top quality via LM-based filtering and LLM refining, particularly beneficial for later training stages requiring higher data quality (Hu et al., 2024). In the code domain, we fine-tuned a small language model to filter math-relevant code snippets at scale, yielding MegaMath-Code. For MegaMath-Synthetic, we extracted and refined QA pairs from math web documents, translated non-Python code snippets into Python, and generated interleaved text-code samples from web content. Together, these efforts form a diverse and scalable math dataset backed by extensive empirical pre-training.

Our contribution can be summarized with following offerings in MegaMath:

1. An open math-focused dataset containing 371B tokens with optimized data curation pipelines, and a variety of data variants to cater to customized demands. (§2.1 - §2.5)
2. A comprehensive set of studies and ablation experiments that rigorously evaluate key design choices in the data accumulation process. (§3.1 - §3.4)
3. Empirical demonstrations including head-to-head comparison with existing math datasets (§3.5, Figure 1), and further training on latest Llama-3 series of models. (§3.6)

## 2 MegaMath Data Curation

In this section, we will describe MegaMath's whole data processing pipelines, which include three main components: web data (§ 2.1), code data (§ 2.2), and synthetic data generated from the former two (§ 2.3). Our key design choices are validated through downstream benchmarks or split validation sets. For computationally intensive operations like deduplication, we prioritize solutions that balance efficiency and effectiveness.

### 2.1 Curating MegaMath-Web

Web data takes up quite a lot of the general pre-training corpora, from which Common Crawl is what has been widely used as pre-training data in many recent LLMs training (Dubey et al., 2024; Yang et al., 2024a; Liu et al., 2024). In MegaMath, we use 99 Common Crawl snapshots (`2014-15` to `2024-46`) as data source to extract high-quality math documents on the Internet. The overall pipeline for web data is presented in Figure 2, with a detailed description in the following subsections. In short, our pipeline contains the following steps: (1) *data acquisition*; (2) *first round text extraction*; (3) `fastText`-*based math filtering*; (4) *deduplication*; (5) *second round text extraction*; (6) *further filtering and post-processing*.

#### 2.1.1 Data Acquisition, URL Filtering, and Language Identification

Instead of using **WET** (**W**ARC **E**ncapsulated **T**ext) data or simply filtering from public pre-training datasets (Penedo et al., 2024; Tang et al., 2024), we re-extracted all the text from

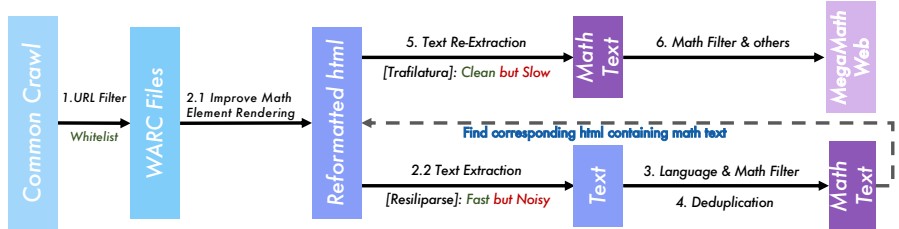

Figure 2: The pipeline for curating MegaMath-Web from Common Crawl data.

**WARC** (**W**eb **ARC**hive format) file format where each web page is stored as an HTML file. This practice enables us to optimize text extraction from HTML, enhancing corpora quality specifically for math domain (§ 2.1.2). We downloaded all available CC dumps and applied a URL filtering strategy before text extraction (Penedo et al., 2023) to exclude domains related to adult, gambling content, etc. Next, we used an off-the-shelf `fastText` model (Joulin et al., 2016) for language identification and retained only English documents (score≥ 0.65).

### 2.1.2   Improved Text Extraction for Math Content

Extracting texts from WARC using common extractors (e.g., `Resiliparse` and `trafilatura`) could produce higher-quality corpora over WET extraction (Li et al., 2024a). However, these extractors often fail to preserve math symbols and equations, even omitting them entirely (Lozhkov et al., 2024a). To address this, we introduced several HTML parsing optimizations specifically for **optimizing math expressions before extraction**. Our approach involved traversing the HTML DOM tree parsed by `Resiliparse` and modifying math elements-related nodes to obtain an improved HTML file easy for text extraction including:

1. **Math Element Conversion:** Converts MathML and KaTeX content into LaTeX by extracting annotation tags or using mathml2latex (with namespace handling) and recursively parsing HTML to accurately extract subscripts and superscripts.

2. **LaTeX Standardization and Transformation:** Removes unnecessary style commands, fixes symbol formatting issues, and converts HTML tags (such as , , and "intbl" spans) into appropriate LaTeX constructs.

3. **Unicode and Entity Conversion:** Maps mathematical Unicode characters and HTML entities to their corresponding LaTeX commands using W3C standards.

**Two-stage Extraction**   Our extraction process consisted of two phases, each serving a distance purpose. In practice, `Resiliparse` and `trafilatura` are widely used for pre-training corpora construction, but they have trade-offs: `Resiliparse` is significantly faster and retains HTML elements more faithfully, while `trafilatura`, though slower, removes noise more aggressively using various extraction engines and heuristics. Unlike prior works based solely on `Resiliparse` (Paster et al., 2024; Han et al., 2024), our pipeline first applied `Resiliparse` for rapid extraction and filtering, significantly shrinking the candidate data size. For these candidate data, we then used `trafilatura` on their WARC files for a second round HTML optimizations and text extraction, obtaining cleaner mathematical data. This coarse-to-fine approach improves text quality while maintaining development efficiency.

### 2.1.3   Robust Math Document Recall

Common Crawl (CC) contains a vast array of texts from diverse domains. To effectively filter texts at scale, we require a robust and efficient classifier. We used `fastText` (Bojanowski et al., 2017), a lightweight n-gram model, to score and identify math-related texts. During development, we identified the following key factors to obtain a robust `fastText` classifier:

1. **Text normalization**: Techniques like tokenization, case folding, digit normalization, and Unicode handling while managing whitespace and special characters achieve better training compatibility.
2. **Seed data**: Uniform sampling from Common Crawl and adding CoT data helps.
3. **Comprehensive evaluation**: Expanding beyond web texts to Wikipedia, textbooks, StackExchange and research papers improves recall assessment.

**fastText Training**  We started `fastText` training with one million positive and negative seed documents from Open-Web-Math and random web documents from CC. Initially, we used a single snapshot dump for development, which risked reinforcing biases. To mitigate this, we sampled from all CC dumps and retrained the classifier during the second-round filtering process. We used Llama-3.1-70B-Instruct (Dubey et al., 2024) to automatically annotate math relevance scores (see Figure 7 for the prompt) on these filtered documents and CoT data was incorporated into the positive set as well, resulting in two million seed data. We used the same training hyperparameters as DeepSeekMath (Shao et al., 2024).

**fastText Evaluation**  When iterating training strategy, we found evaluation on 20K in-distribution (ID) samples yielded easily **over 90%** F1 score, masking `fastText`'s true performance. We thus created an out-of-distribution (OOD) suite by sampling arXiv, Stack-Exchange, Wikipedia, and Textbook data from MathPile (Wang et al., 2024). In the OOD setting, our text normalization and training adjustments boosted the average F1 score from **81.8%** to **98.8%**, validating the effectiveness of our training strategy.

### 2.1.4  Data Deduplication

Data deduplication plays a vital role in data curation process, especially for improving training efficiency, stability and reducing data memorization (Lee et al., 2022; Tokpanov et al., 2024). We adopted the Locality Sensitive Hash (LSH) implementation of MinHash (Broder, 2000) for efficiency. Given two documents, the probability that they are assigned to the same hash bucket depends on their Jaccard similarity (Broder, 1997) $S$ and is given by $P = 1 - (1 - S^b)^r$ where $b$ denotes the number of hash functions per bucket and $r$ represents the number of buckets. Given a fixed hash permutation scheme ($b \times r$), which is strongly correlated with memory cost, and a target Jaccard similarity threshold $t$, it is desirable to find the optimal deduplication configuration—one that ensures a rapid decay of $P$ for any $S \leq t$. Considering our CPU capacity, we evaluated multiple configurations with the number of permutations between 110 and 128 and $t \in \{0.70, 0.75, 0.80\}$. Assisted by training experiments, we determined that the most feasible choice is $r = 11$, $b = 10$, and $t = 0.75$.

### 2.1.5  Curating MegaMath-Web-Pro: A Premium Subset

It is increasingly common practice to filter top-quality data due to its superior impact on model performance (Abdin et al., 2024). High-quality data not only enhances performance but also does so at a lower cost, making it ideal for continual pre-training, mid-training, or scenarios with limited budgets. We thus further developed MegaMath-Web-Pro, a premium subset filtered and refined from MegaMath-Web. We employed the FineMath classifier (Lozhkov et al., 2024a) to filter out low-quality text. Subsequently, we used LLMs to further refine the text, **ultimately delivering 15.1B tokens that significantly surpass all existing math corpora such as FineMath-4plus (Lozhkov et al., 2024a) (cf. Figure 1)**. Though LLM was involved, we focused primarily on noise removal and text reorganizing thus this is not categorized as pure synthetic data. See §B.2 for full developing strategy.

## 2.2  Curating MegaMath-Code

Code pre-training has proved to enhance general reasoning (Shao et al., 2024; Aryabumi et al., 2024), and LLMs have also shown great potential to leverage code for problem-solving (Gou et al., 2024; Li et al., 2024b). Thus, we believe blending code in math LLM training is also crucial. We built MegaMath-Code based on the Stack V2 (Lozhkov et al., 2024b), and employed a multi-step pipeline to recall high-quality code relevant to mathematical reasoning, logic puzzles, and scientific computation. As shown in Figure 3, our pipeline consists of: (1) *Programming Language Selection* (§2.2.1) and (2) *SLM-based Code Recall* (§2.2.2).

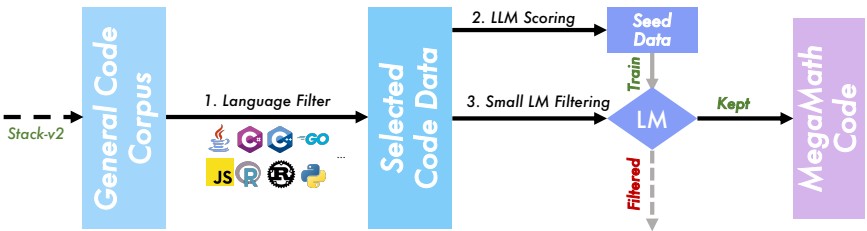

Figure 3: The pipeline for curating MegaMath-Code.

### 2.2.1 Programming Language Selection

The code pre-training corpus includes hundreds of programming languages; however, many of these languages are primarily associated with domains that are not closely related to mathematics or scientific computation (Lozhkov et al., 2024b). In order to reduce the cost of model-based recall, we selected eleven programming languages based on choices made in previous studies (Azerbayev et al., 2023; Xu et al., 2024b): C, C#, C++, Go, Java, JavaScript, Python, R, Rust, Shell, SQL. The selected languages are either extensively used in scientific computing and numerical operations or represent a significant portion of the corpus, which may potentially include mathematics-related snippets.

### 2.2.2 SLM-based Code Data Recall

We applied a small language model (SLM) based recall mechanism to identify math-related code snippets from public code pre-training datasets. Inspired by recent works (Penedo et al., 2024; Zhou et al., 2024; Wei et al., 2024), we first used a strong LLM to score code quality (educational value) and mathematical relevance, assigning a discrete score from 0 to 5 for each aspect, as applied in several works (Yuan et al., 2024; Penedo et al., 2024). Then, we trained a SLM on these data for large-scale filtering. (Please see §C for more details). We also found that: (1) **Stricter filtering greatly enhances performance to solve problems using code**; (2) **Allocating no more than 20% of code data maximizes code-integrated problem-solving ability while maintaining NL reasoning benefits**. This aligns with DeepSeekMath's (Shao et al., 2024) training recipe and further reinforces the justification for our filtering strategy, and we also empirically show the reasonability of this choice in §3.3.

### 2.3 Curating MegaMath-Synthetic Data

Beyond being a high-quality mathematical corpus, MegaMath also serves as **a strong foundation for large-scale data synthesis**. We explored data synthesis methods to further enhance both the quantity and quality of our dataset. Our synthesis spans three distinct formats: (1) *Q&A data*, (2) *code data*, and (3) *text & code block data*.

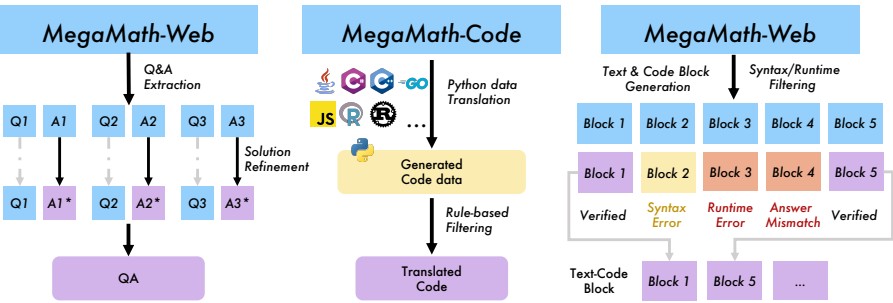

Figure 4: The pipeline for curating synthetic data. **Left**: QA data generation; **Middle**: Python code augmentation; **Right**: text & ode block data curation.

**Q&A Extraction** Question-and-answer data is inherently well-structured and embodies a concentrated form of knowledge, making it valuable for problem-solving benchmarks (Maini et al., 2024). Recent work reveal that these data can be found in pre-training data with massive quantity (Yue et al., 2024). We thus integrate and further verify this in

MegaMath. Our pipeline contains two steps: (1) identify and extract Q&A pairs from the raw documents; (2) refine the Q&A to make up or improve the intermediate reasoning steps. To improve diversity and accumulate quantity, we ensembled refined Q&A data from Qwen-2.5-72B-Instruct (Yang et al., 2024a) and Llama-3.3-70B-Instruct (Dubey et al., 2024).

**Code Translation**  To enhance Python code data, we employed LLMs to "translate" code from other programming languages into Python, thereby augmenting the code data. We adopted a straightforward zero-shot prompting approach using open-source LLMs or code-specialized LLMs to enhance the volume of Python code. Specifically, we experimented with two models: Qwen2.5-Coder-32B-Instruct (Hui et al., 2024) and Llama-3.1-70B-Instruct (Dubey et al., 2024). Please kindly refer to §C for the translation prompt.

**Text & Code Block Generation**  Recent work by Lu et al. (2024) introduced a synthesis pipeline for obtaining *"generated mathematical code"*, consisting of interleaved text, symbolic expressions and code blocks. Such data has been shown to enhance a model's ability to generate Python snippets for solving mathematical problems and to leverage execution feedback to refine solution steps. As illustrated in Figure 4, we unify such process into: (1) LLM-based generation: given a document, LLMs generate multiple structured blocks including title, mathematical expression, result, and corresponding code; (2) Verification via execution: ensures that the generated code executes correctly without errors and produces expected outputs; (3) Packing verified blocks: combines validated blocks into a single training sample for downstream use. Besides, we found Lu et al. (2024) did not account for malicious code, handling only basic errors like timeouts. To address these issues, we implemented a pre-filtering mechanism based on Abstract Syntax Tree (AST). Any snippet flagged as risky is excluded from execution, safeguarding a 100% execution success rate without abrupt halts or segmentation faults during our curation.

## 2.4  Dataset Decontamination

To mitigate benchmark contamination (Xu et al., 2024a), we checked the overlap between MegaMath and 12 downstream benchmarks widely used in evaluating LLM's mathematical reasoning ability such as GSM8K (Cobbe et al., 2021), MATH (Hendrycks et al., 2021), MMLU (Hendrycks et al., 2020), and AIME [1]. We concatenated problems and solutions together as a whole sample, checked the exact 13-gram match, and ruled out contaminated documents. This further removes about **0.01%** of the documents from the dataset.

## 2.5  The Final Dataset: MegaMath 371B Collection

Combining all previous efforts together, the final collection of MegaMath datasets currently contained a total of 371B tokens (count by the Llama-2 tokenizer). We present a detailed breakdown statistics about MegaMath in **Table 1**. Designed for various training stages, training budgets, and base model capability, we offer a collection of MegaMath data

Table 1: The category and statistics of MegaMath.

| Category | # Sample(M) | # Toks(B) | Avg. (# Toks) |
|---|---|---|---|
| **Web Domain** | **121.5** | **279.0** | **2296.9** |
| Web | 106.5 | 263.9 | 2478.7 |
| Web-Pro | 15.0 | 15.1 | 1006.0 |
| **Code Domain** | **13.4** | **28.1** | **2102.7** |
| **Synthetic Data** | **80.2** | **64.5** | **804.5** |
| Translated Code | 7.4 | 7.2 | 979.5 |
| Q&A | 22.6 | 7.0 | 308.3 |
| Text&Code Block | 50.2 | 50.3 | 1002.1 |
| **Total** | **215.1** | **371.6** | **1727.6** |

variants including: (1) **MegaMath-Web**: the complete web dataset consisting of 263.9B tokens, and also **MegaMath-Web-Pro** (15.1B), the top-quality subset obtained through LM-based scoring and refining. (2) **MegaMath-Code** (28.1B): math-related code corpus recalled from Stack-v2. (3) **MegaMath-Synth** (64.5B): LLM-based synthetic data enhancing both the quality and quantity, covering three distinct formats of text and code data.

---

[1]https://huggingface.co/datasets/AI-MO/aimo-validation-aime

# 3 Ablation and Demonstration of MegaMath at Scale with Pre-training

During data curation, we conducted extensive pre-training experiments on MegaMath to ablate each key decision. In this section, we present the experimental details, key results, and finally scale up training to further demonstrate the effectiveness of MegaMath.

## 3.1 Setup

**Proxy LM for Ablation** During development, we used a small proxy model for ablations on each data source and component. We chose TinyLlama-1B (Zhang et al., 2024) for its small size and transparent training, ensuring it effectively monitors data quality. We trained within a controlled budget, typically set to **5/15/55** B tokens, depending on dataset size and experimental cost, and evaluated performance at 1B token intervals.

**Evaluation** We used a total of 10 math-related benchmarks, splited into two sets: **Core** and **Extended**. The **Core** set includes five math-focused tasks with stable improvements even under limited training, such as GSM8K and MATH. Building on this, the **Extend** set further includes five datasets, either indirectly related to math or with performance fluctuations, such as MMLU-STEM. We employ two prompting-based evaluations: (1) few-shot CoT (Wei et al., 2022) for all benchmarks; (2) PAL (Gao et al., 2023) for the **Core** set to assess problem-solving via Python code generation. Please check § G for more details.

## 3.2 Ablation on MegaMath-Web

**Importance of optimizing text extraction for math content** We conducted continual pre-training experiments within a 15B-token training budget on one dump from 2024. The training corpora consisted of filtered math documents from vanilla `trafilatura`, and text extracted from the optimized HTML using `Resiliparse` and `trafilatura`, all derived from the first-round filtering. During original `trafilatura`'s extraction, `<math>` elements in HTML were directly discarded. After applying specialized optimizations for math-related HTML, the extracted data from `trafilatura` well-preserved math symbols and clearly improved CoT downstream performance (cf. Table 2). When both extractors operated on our optimized HTML, `Resiliparse` preserved more noise from the original documents, leading to lower data quality compared to `trafilatura`.

Table 2: Ablation on Text Extraction for Math

| Text Extractors | w/ HTML Optimization | Core Avg. | Ext. Avg. |
|---|---|---|---|
| *Base Model* | - | 11.2 | 14.7 |
| trafilatura | ✗ | 22.0 | 19.2 |
| Resiliparse | ✔ | 22.5 | 18.6 |
| trafilatura | ✔ | **23.8** | **20.6** |

**Parameters of Deduplication** To minimize redundancy and reduce the costs associated with follow-up text re-extraction using `trafilatura`, we conducted ablation pre-training experiments on all 2014 dumps with a 55B-token training budget to optimize the parameters for Minhash LSH. Our goal was to preserve downstream CoT performance while retaining as many mathematical documents as possible within our cluster capacity. As shown in Table 3, applying $r = 11$, $b = 10$ provided the optimal balance. We reported the average of the last 5 checkpoints to avoid result fluctuations.

Table 3: Ablation on MinhashLSH Dedup.

| (r, b) | t | Tokens Left (B) | Core Avg. | Ext. Avg. |
|---|---|---|---|---|
| (14, 9) | 0.70 | 16.0 | 17.3 | 16.6 |
| (14, 8) | 0.75 | 23.5 | 19.1 | 17.0 |
| **(11, 10)** | **0.75** | **26.0** | **19.4** | **17.5** |
| (11, 11) | 0.75 | 25.0 | 19.2 | 16.0 |
| (9, 12) | 0.80 | 29.0 | 18.8 | 16.9 |
| (9, 13) | 0.80 | 30.0 | 17.6 | 15.7 |

**Ablation on `fastText`** Initially, we employed Open-Web-Math as the positive seed data for training `fastText` used for the first round filtering with a loose threshold. While it worked well on the single dump used for initial development, it became less accurate when scaled to all dumps, likely due to shifting data distributions. We thus re-trained `fastText` with LLM-annotated math-related documents from all dumps as the positive seed data. As shown in Figure 5, the re-trained `fastText` (V2) performed better in the second-round filtering compared to the initial version (V1). Our controlled experiments revealed that balanced sampling seed data from each dump provides a slight improvement while incorporating

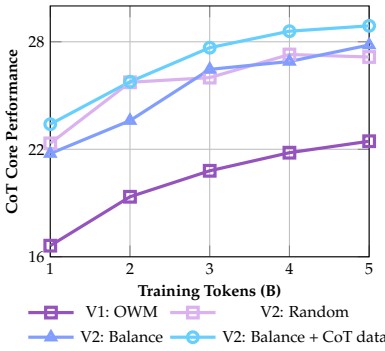

Figure 5: Ablation on `fastText`

CoT data into positive seed data yielded significant gains. Note that the ablations were conducted on the top 10% scoring filtered data from all dumps in 2024 within a 5B-token training budget. We further validated our decision through experiments on all dumps yearly as shown in Figure 9.

## 3.3 Ablation on MegaMath-Code

We conducted two sets of ablation studies using 5B training tokens: (1) evaluating the impact of recall filtering criteria on downstream performance [2], and (2) ensuring the recalled code dataset is sufficiently large for training despite aggressive filtering. As shown in Table 4, stricter filtering (i.e., $S_{edu} \geq 4$ and $S_{math} \geq 4$) selects code data that significantly boosts PAL performance. Moreover, even though such strict filtering excludes many samples, the remaining data appears sufficient for effective mathematical training; in fact, mixing too much code data seems harmful for CoT Performance (*e.g.*, $\geq 30\%$).

Table 4: Ablation of filtering criteria.

| Filter Criteria | CoT Avg. | PAL Avg. |
|---|---|---|
| **text only** | 19.0 | 15.6 |
| $S_{edu} \geq 3, S_{math} \geq 3$ | 19.4 | 16.1 |
| $S_{edu} \geq 3, S_{math} \geq 4$ | 19.8 | 16.8 |
| $S_{edu} \geq 4, S_{math} \geq 3$ | 19.7 | 17.5 |
| $S_{edu} \geq 4, S_{math} \geq 4$ | 18.8 | 19.5 |

Table 5: Ablation of data mixture ratios.

| Mix Ratio | CoT Avg. | PAL Avg. |
|---|---|---|
| **text only** | 19.0 | 15.6 |
| **code : text** = 12.5% | 19.3 | 17.4 |
| **code : text** = 20.0% | 19.5 | 18.4 |
| **code : text** = 33.3% | 16.4 | 17.5 |
| **code : text** = 50.0% | 17.5 | 18.8 |

## 3.4 Ablation on MegaMath-Synthesis

Our synthesis development proceeds in parallel with web data acquisition. Thus, in web data synthesis, we started with existing public corpora rather than MegaMath-Web. In particular, we utilized quality-filtered subsets of Open-Web-Math and Infimm-Web-Math to lower experimental cost. We focus on: (1) Verify that the generated data can boost performance more effectively. (2) Evaluate the impact of different prompts and models on performance, ultimately guiding better strategies.

**QA generation** We implemented the two-stage pipeline in WebInstruct (Yue et al., 2024) but using the latest and most capable LLMs. Through several prompting iterations, we found: (1) using an ELI5-style (Explain like I am five) prompt for QA refining produces structured solutions; (2) emphasizing information completeness enhances data quality further. Table 6 shows the 5B training results on different datasets: using

Table 6: Ablations on prompt and comparison with other data. **FM-4plus**: FineMath-4plus.

| Data | Core Avg. | Ext. Avg. |
|---|---|---|
| **FM-4plus** | 28.3 | 19.6 |
| **WebInstruct** | 34.6 | 17.6 |
| **Vanilla Prompt** | 39.2 | 19.5 |
| *w.* ELI5 | 41.3 | 19.2 |
| *w.* ELI5 + IC | **48.8** | **23.6** |
| **MegaMath-Synth-QA** | **49.6** | **23.8** |

prompt with ELI5 improves performance, and further adding information completeness

---

[2]When testing filtering criteria, we prioritize PAL results, and use Python subset for training.

(ELI5 + IC) yields the best ablation results, with `Core` and `Extended` scores of **48.8** and **23.6**. Besides, we also show the performance on our final MegaMath-Synth-QA, demonstrating overall best results. These results indicate that structured and comprehensive extraction are keys to enhancing QA data, and also show that QA style data exhibits superior performance to web documents.

**Code Synthesis** Our experiments compared training on raw code data to methods incorporating code translation and interleaved text & code blocks. With a controlled training budget of 5B tokens (see Table 7), translated code (trans. code) yields modest downstream improvements over raw code, while adding code block data further enhances both CoT and PAL performance, even without mixing text data (see the "full" line

Table 7: Ablations on Code Synthesis: The default code-to-text ratio is 1:7; "full": no text is mixed. We exclude Lu et al. (2024) due to its partial release (≈0.25B tokens).

| Data | CoT Avg. | PAL Avg. |
|---|---|---|
| code | 18.8 | 19.5 |
| trans. code | 19.0 | 20.6 |
| text & code block | 22.5 | 28.1 |
| text & code block (full) | **30.8** | **46.5** |
| MegaMath-Text & Code Block (full) | **33.5** | **46.3** |

results). We also report results on our final MegaMath-Text & Code Block corpus. Overall, these results clearly show that synthetic data achieves higher quality.

## 3.5 Comparison with Existing Math Corpora

To assess the data quality of MegaMath, we performed continual pre-training on existing corpora within a 55B token budget. We compared MegaMath-Web with mainstream large-scale corpora, including Open-Web-Math, Infimm-Web-Math, and the latest FineMath release. As previously shown in **Figure 1**, MegaMath-Web already achieves corpus quality comparable to Infimm-Web-Math in downstream tasks but providing substantial more tokens, with performance improving if we use higher-scored subsets (top 75% and top 50%). Notably, MegaMath-Web-Pro outperforms both FineMath-3+ and FineMath-4+ by $\geq$ **4**%, delivering the highest-quality corpus to date. Furthermore, these MegaMath-Web variants show the potential to offer flexible options to accommodate different computing budgets.

## 3.6 Putting It All Together: Training MegaMath on Cutting-Edge LMs

We demonstrate the effectiveness of MegaMath by training it on state-of-the-art open LLMs—the Llama-3.2 series. Given that the Llama-3 models have been extensively trained on 14.8T tokens (Dubey et al., 2024) and exhibit strong performance across various benchmarks, we believe they exemplify state-of-the-art capability and robust performance, making them an ideal validation point. For training, we adopt and refine the data mixture configurations from DeepSeek-Math and Llemma to accommodate our diverse data sources, and train 100B tokens for Llama-3.2-1B and 50B tokens for Llama-3.2-3B. We evaluate all models under CoT and PAL configurations. As shown in Figure 6, the MegaMath series of models achieves a 15% to 20% CoT performance improvement over Llama — for example, reaching 56.2% on GSM8K and 25.1% on MATH for the 3B model — with a similar boost observed on PAL. This clearly demonstrates the exceptional quality and effectiveness of MegaMath in advancing mathematical reasoning in state-of-the-art language models. Please refer to § F.2 and § G.2 for training configuration and full evaluation results.

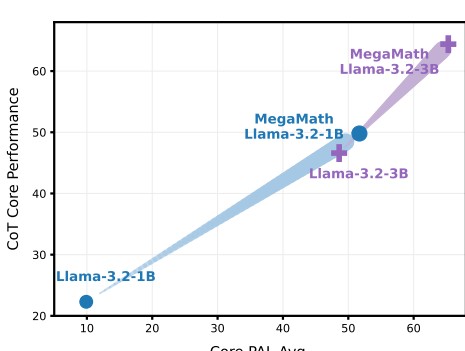

Figure 6: Training on Llama-3.2-1B/3B.

## 3.7 Pre-training on MegaMath Could also Boost Post-training

A key question is whether the performance boost from pre-training on MegaMath endures after the model undergoes supervised fine-tuning (SFT). To investigate this, we conducted SFT on both the standard Llama-3.2-3B-Base and our MegaMath-Llama-3.2-3B-Base.

For this process, we used a random 5M subset of the OpenMathInstruct-2 dataset. To ensure a fair comparison, both models were trained for one epoch with identical hyperparameters: a learning rate of 5×e-6, a global batch size of 256, training for one epoch and the same zero-shot evaluation protocol.

The results in Table 8 show that our MegaMath-pretrained model consistently outperforms the baseline Llama-3.2-3B-Base after SFT. Notably, it also achieves significantly stronger performance than the official Llama-3.2-3B-Instruct model across all math benchmarks. This demonstrates that specialized mathematical pre-training on MegaMath provides a durable advantage that is amplified, not overwritten, by subsequent post-training.

Table 8: Performance gains from pre-training on MegaMath persist after SFT. We also report the performance of the official Llama-3.2-Instruct for reference.

| Model | ASDIV | GSM8K | MATH500 | MATH | MAWPS | OCW | SVAMP | TabMWP |
|---|---|---|---|---|---|---|---|---|
| Llama-3.2-3B-Instruct | 78.1 | 67.5 | 32.6 | 32.8 | 90.4 | 7.7 | 81.4 | 37.6 |
| Llama-3.2-3B-Base + SFT | 85.8 | 73.1 | 41.4 | 45.6 | 93.1 | 7.0 | 79.8 | 65.9 |
| MegaMath-Llama-3.2-Base + SFT | **91.3** | **82.7** | **52.6** | **55.1** | **95.1** | **13.6** | **86.2** | **73.6** |

## 4 Related Works

**Mathematical Pre-training Corpus and Syntheic Datasets** OpenWebMath (Paster et al., 2024) curated its data from web pages, with strict filtering which may remove potential documents. MathPile (Wang et al., 2024) diversified from web domains and built datasets mostly from arXiv papers and textbooks. Furthermore, InfiMM-Web-Math (Han et al., 2024) assembled a multimodal dataset pairing math text with images. Recently, FineMath (Lozhkov et al., 2024a) was developed by retrieving from FineWeb (Penedo et al., 2024) and using a BERT classifier to select clear, step-by-step math explanations. For synthetic math datasets, recent work such as NuminaMath (Li et al., 2024b) converted competition-level problems into chain-of-thought solutions via tool-assisted reasoning. Meanwhile, Skywork-Math (Zeng et al., 2024), OpenMathInstruct-2 (Toshniwal et al., 2024) and WebInstruct (Yue et al., 2024) generated large-scale QA pairs from open benchmarks and web contents. MathCoder2 (Lu et al., 2024) used a 19.2B-token MathCode-Pile combining filtered datasets with synthetic code data. In MegaMath, we aim to build a large-scale dataset that matches proprietary corpora via reproducible pipelines, diverse data sources, and thorough sanity checks, finally covering larger quantity and higher quality dataset.

## 5 Conclusion and Future Work

We introduce MegaMath, the largest training corpus to date tailored for the mathematical domain, comprising 371B tokens from web sources, code corpora, and synthesized data. Comprehensive ablation studies guide us to efficient curation of high-quality, domain-specific datasets. Large-scale continual pretraining on Llama-3 series of model further demonstrates MegaMath's effectiveness by producing strong math base models. We hope the MegaMath dataset, alongside our released artifacts, can foster further research in mathematical reasoning and domain-specific language modeling.

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

# A Comparison with Existing Corpora

Table 9: Comparison with existing large-scale math corpora

| Corpus Name | Fully Public | # Tokens (B) | Date | Type |
|---|---|---|---|---|
| OpenWebMath | ✔ | 14.5 | 2023 Oct. | Web |
| AlgebraicStack | ✔ | 11.0 | 2023 Oct. | Code |
| MathPile | ✔ | 9.5 | 2023 Dec. | ArXiv, Web, Textbooks, StackExchange, Wiki |
| DeepseekMath | ✗ | 120.0 | 2024 Feb. | Web |
| InfiMMWebMath | ✔ | 55.0 | 2024 Sep. | Web |
| Qwen Math Corpus v2 | ✗ | 1000.0 | 2024 Sep. | Web, Code snippets, Encyclopedias, Books, Exam questions, Synthetic data |
| MathCode-Pile | ✗ | 19.1 | 2024 Oct. | Web, Code, Textbooks |
| **MegaMath Collection (Ours)** | | | | |
| MegaMath-Web | ✔ | 263.9 | | Web |
| MegaMath-Web-Pro | ✔ | 15.1 | | Web |
| MegaMath-Code | ✔ | 28.1 | 2025 | Code |
| MegaMath-Synth-Code | ✔ | 7.2 | | Code |
| MegaMath-Synth-Q&A | ✔ | 7.0 | | Q&A |
| MegaMath-Synth-Text&Code | ✔ | 50.3 | | Interleave text&code |

# B Details for Curating MegaMath-Web

The scoring prompt for evaluating web documents' relevance to mathematics is presented in Figure 7.

```
Please evaluate the given document for its relevance to mathematics and assign a
score from 0 to 5. Use the following scoring criteria:
5: The document is entirely about mathematics, containing numerous mathematical
concepts, formulas, proofs, or advanced mathematical educational content.
4: The document is primarily about mathematics but may include some applications
in other disciplines or content related to mathematics education.
3: The document contains significant mathematical content, but it's not the main
focus. It might be mathematical applications in physics, engineering, or similar
fields.
2: The document includes some mathematical elements, such as basic calculations,
simple statistics, or graphs, but these are not the main content of the document.
1: The document has very little mathematics-related content, possibly only
mentioning numbers or simple calculations in passing.
0: The document has no mathematical content whatsoever.

The document is given as:
<EXAMPLE>.

After examining the document:
- Briefly justify your total score, up to 100 words.
- Conclude with the score using the format: "Score:  <total points>"
```

Figure 7: Scoring Prompts for evaluating web documents relavance to mathematics.

## B.1 Fine-grained Deduplication

We also explored several fine-grained deduplication methods, including exact substring (Lee et al., 2022) and sentence-level deduplication (Raffel et al., 2020). Initially, we found that removing duplicates disrupted text consistency. To mitigate this, we attempted trimming only the head and tail portions, but still identified many math expressions and degraded downstream performance. We suspect the effectiveness of these methods depends on text

extraction techniques and may be more suitable for `Resiliparse`. We thus leave this for future exploration.

## B.2 Strategy for MegaMath-Pro Subset

Building on Gunasekar et al. (2023), documents with higher educational values are treated as higher-quality samples—a strategy widely adopted in pre-training works. In MegaMath, we create the MegaMath-Web-Pro subset from MegaMath-Web data using FineMath classifier (Allal et al., 2025) to score documents on a 0–5 scale. However, we found that document distribution and relevance to mathematical reasoning vary over time. As Table 10

Table 10: Yearly Ablation of Edu scoring strategy.

| | GSM8K | MATH | ASDiv | SVAMP | MAWPS | AVG |
|---|---|---|---|---|---|---|
| **FM-4plus** | 10.5 | 6.1 | 41.9 | 25.3 | 57.9 | 28.3 |
| **2014** | 6.0 | 3.7 | 30.2 | 17.5 | 35.4 | 18.6 |
| **2015** | 5.0 | 3.0 | 21.8 | 14.4 | 27.7 | 14.4 |
| **2016** | 3.9 | 4.4 | 28.4 | 16.9 | 35.8 | 17.9 |
| **2017** | 6.4 | 5.0 | 34.9 | 21.8 | 44.6 | 22.5 |
| **2018** | 6.2 | 5.9 | 34.6 | 22.6 | 46.7 | 23.2 |
| **2019** | 6.4 | 4.8 | 37.7 | 21.7 | 48.4 | 23.8 |
| **2020** | 8.7 | 4.6 | 35.5 | 24.7 | 49.3 | 24.6 |
| **2021** | 8.3 | 5.2 | 39.6 | 24.6 | 53.0 | 26.2 |
| **2022** | 10.5 | 5.3 | 41.9 | 24.4 | 56.4 | 27.7 |
| **2023** | 12.1 | 5.8 | 45.2 | 28.3 | 63.0 | 30.9 |
| **2024** | 14.4 | 6.1 | 46.6 | 28.6 | 63.9 | 31.9 |

indicates, after applying Edu filtering, training 5B tokens on some years' data (e.g., 2014) yields marginal improvements, whereas later years achieve much higher performance than FineMath-4plus (FM-4plus). Based on these observations, we adopted a dynamic filtering strategy: a more tolerant threshold (Edu score $\geq$ 3) for recent years (e.g., 2023–2024) and a stricter one (Edu score $\geq$ 4) for earlier periods (e.g., 2014–2017). Similar to Nemontron-CC (Su et al., 2024), we also used an LLM (in our case, we use Llama-3.3-70B-instruct) to further remove noise, and refine the web text into higher quality. Please see Figure 8 for the detailed prompt.

```
Task:
- Carefully analyze the provided text to extract key facts, concrete details,
important numbers, and core concepts.
- Remove any irrelevant or noisy information, and reorganize the content into a
logically structured, information-dense, and concise version that is easy to
learn from. Output only the refined text.
- Strive to maintain the original length as much as possible (avoid excessive
shortening).

Text:
<EXAMPLE>

Just output the refined text, no other text.
```

Figure 8: Rewriting Prompt for constructing MegaMath-Web-Pro.

## B.3 Further Ablation on `fastText`

We further validated our decision through experiments on all yearly dumps. Specifically, we conducted pre-training on the top 10% highest-scoring filtered data from each yearly dump using different versions of `fastText`. As shown in Figure 9, the results confirm the effectiveness of our final version (V2: Balance + CoT data), demonstrating clear improvements over the initial version in our second-round filtering. Another interesting observation is that data quality, as indicated by downstream performance, gradually improves over time.

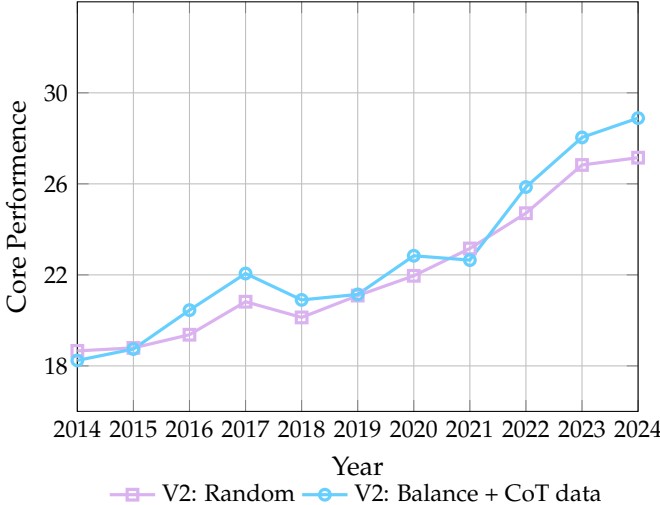

Figure 9: Ablation on `fastText` for each year's all dumps within 5B-token training budget

## C  Details for Curating MegaMath-Code

We used Llama-3.1-70B-Instruct to annotate 25K randomly sampled code data and fine-tuned a Qwen-2.5-0.5B model to judge the code quality and decide whether to filter the code. The scoring prompts are presented in Figure 10 and Figure 11. We only keep code data with Math Score $\geq$ **4** and Code Score $\geq$ **4**, and all other code data are treated as negative samples during training.

In Table 11, we list details for our supervised fine-tuning configurations. We use LlamaFactory (Zheng et al., 2024) as our code base. Same as ProX (Zhou et al., 2024), we also select the model with highest F1 score as out final recalling models, which achieves 80% on a split validation set.

Table 11: Training parameters for SLM.

| HyperParams | Setting |
|---|---|
| LR | 1e-5 |
| LR Schedule | cosine |
| Batch Size | 64 |
| Number of Epochs | 2 |
| Context Length | 2048 |

```
Below is an extract from a resource focused on mathematical reasoning. Evaluate
its educational value in effectively teaching concepts in this area, with emphasis
on mathematical reasoning. Use the additive 5-point scoring system described
below. Points accumulate based on each criterion:

- Add 1 point if the resource contains valid content in mathematics, reasoning,
logic puzzles, or scientific computation, even if it's not inherently educational
(e.g., configurations or specialized algorithms).
- Add another point if the resource addresses practical concepts in these areas,
such as solving math problems or reasoning tasks, even without annotations or
explanations.
- Award a third point if the resource is suitable for educational use and
introduces key concepts in mathematics or reasoning, with a structured format and
some explanations or annotations.
- Give a fourth point if the resource is self-contained and directly useful for
teaching, resembling a structured exercise, tutorial, or part of a lesson in
mathematical reasoning or logic.
- Grant a fifth point if the resource is outstanding in educational value and
perfectly suited for teaching, with clear, step-by-step explanations and thorough
annotations on mathematical reasoning concepts.

The extract: <EXAMPLE>

After examining the extract:

- Briefly justify your total score, up to 100 words.
- Conclude with the score using the format: "Score: <total points>"
```

Figure 10: Scoring Prompt for evaluating code snippets' relevance to mathematics.

```
Below is an extract from a <CODE_TYPE> program. Evaluate whether it has a high
educational value and could help teach coding. Use the additive 5-point scoring
system described below. Points are accumulated based on the satisfaction of each
criterion:

- Add 1 point if the program contains valid <CODE_TYPE> code, even if it's not
educational, like boilerplate code, configs, and niche concepts.
- Add another point if the program addresses practical concepts, even if it lacks
comments.
- Award a third point if the program is suitable for educational use and
introduces key concepts in programming, even if the topic is advanced (e.g., deep
learning). The code should be well-structured and contain some comments.
- Give a fourth point if the program is self-contained and highly relevant to
teaching programming. It should be similar to a school exercise, a tutorial, or a
<CODE_TYPE> course section.
- Grant a fifth point if the program is outstanding in its educational value and
is perfectly suited for teaching programming. It should be well-written, easy to
understand, and contain step-by-step explanations and comments.

The extract:
<EXAMPLE>

After examining the extract:
- Briefly justify your total score, up to 100 words.
- Conclude with the score using the format: "Score: <total points>"
```

Figure 11: Scoring Prompt for evaluating code snippets' general quality, i.e., educational value.

# D   Details for Curating MegaMath-Synth

**Synthetic Text Data**    We provide the prompts for extraction and refining Q&A below.

```
Below is a web document extract. Assess whether it contains a mathematical
question-and-answer pair:

- If the web document extract does not contain a mathematical question-and-answer
pair, return the explicit symbol `[NO QA]`.
- If a mathematical question-and-answer pair is found, extract it in the following
format:
  Question: <question text with complete problem statement and all necessary
  mathematical information>
  Answer: <complete solution with all necessary steps and calculations included>
  (only if an answer is provided, otherwise do not generate this line)
- The extracted pair must be self-contained and mathematically precise, allowing
independent solving without additional context.

#### The extract:
<EXAMPLE>

Now process the extract and return the result.
```

Figure 12: Prompt for QA extraction.

```
Below is a mathematical question-and-answer pair. Refine the answer based on the
following requirements:

- **If the answer does not contain any explanation or intermediate reasoning
process**:
  - Add only necessary intermediate reasoning process leading to the given answer
  - Ensure the added steps are logical, clear, and provide necessary explanation
  of the solution process

- **If the answer already includes necessary solution process**:
  - Reorganize the solution into a clear and well-structured format for better
  readability and understanding
  - for simple solutions, there is no need to use latex format

- Maintain the original question text and provide the refined answer in the same
format:
  - Question: <question text>
  - Answer: <refined solution>

#### The question-and-answer pair:
<EXAMPLE>

Suppose you are a math teacher, you should explain the solution in a way that is
easy for a student to understand. Now process the pair and return the refined
result.
```

Figure 13: Prompt for Refined QA.

**Synthetic Code Data**  We provide the prompt for code translation in Figure 14 at below.

```
Below is an extract from a code snippet. Translate the code from other programming
languages into Python. Read the code carefully and translate it into Python.

- If the original code has poor quality or cannot be converted to Python, return
the explicit symbol "[Untranslatable]".
- The translated Python code should meet the following requirements:
    - Ensure good code formatting.
    - Include proper comments or explanations for clarity explaining the logic
    where needed.
    - Add docstrings when necessary to improve readability.
    - Wrap the generated Python code within python ```python ```.
    - Keep good test cases if any.

The extract:
```
<EXAMPLE>
```

Do not produce any additional commentary or text beyond ```python ```.
Now output the translated Python code:
```

Figure 14: Prompt for translating non-Python code samples into Python code samples.

**Synthetic Code Block Data**  We used the same prompts as in Lu et al. (2024). Please see Figure 15. Our AST filtering mainly contains the following aspects:

1. **Code Parsing and AST Generation**: The input code is parsed into an AST using Python's built-in `ast` module. The system first verifies code length constraints (max 100,000 characters) and handles syntax errors through exception catching.

2. **Import Declaration Analysis**: A specialized visitor collects all imported modules and their aliases through two-phase inspection:
   - Direct imports (`import x as y`) mapping
   - Selective imports from modules (`from a import b as c`)

3. **Semantic Node Traversal**: A secondary visitor examines all function calls and context managers, checking against three prohibition categories:
   - *File Operations*: file I/O methods (e.g., `open`, `savefig`), path manipulations, and serialization functions
   - *Concurrency Patterns*: Thread/process creation calls and 5+ restricted modules (e.g., `threading`, `asyncio`)
   - *Network Communication*: network libraries and protocol-specific methods (e.g., `requests.get`, `socket.send`)

4. **Module Dependency Verification**: Cross-references imported modules against prohibited libraries spanning file systems (`shutil`), parallelism (`multiprocessing`), and network protocols (`ftplib`).

5. **Context-Specific Checks**: Special handling for:
   - `with` statements containing file open operations
   - Class instantiations of thread/process primitives
   - Path manipulation methods in object-oriented interfaces

```
You will be presented with a text related to math. I need you to identify all the
complex computations in it. For each complex computation that requires a
scratchpad, find out the conditions needed for the computation, the latex
expression that conducts the computation, and the result of the computation. Then
generate a Python code snippet for each computation that demonstrates how the
result is reached. Output each computation in the following format:

Conditions Needed:
1. [Condition 1]
2. [Condition 2]
...

Computation Expression:
$[Latex Expression]$

Computation Result:
[Computation Result]

Python Code Snippet:
```python
[Python Code]
```

There can be more than one complex computation in the text. Output only the
computations that requires calculation. Do not include mathematical statements or
definitions as a computation. Make sure each snippet can be executed individually.
The text is as follows:

<EXAMPLE>

The computations are:
```

Figure 15: Prompt for generating code-block data (Lu et al., 2024).

Table 12: LLM Usage Summary During Data Curation

| Purpose / Stage | LLM Used | Description / Notes |
| --- | --- | --- |
| Math-related web document classification | Llama-3.1-70B-Instruct | Used to annotate seed documents for training FastText classifiers |
| Web document refinement | Llama-3.3-70B-Instruct | Used to further remove noise and refine web text into higher-quality content, similar to Nemotron-CC. |
| Math-related code classification | Llama-3.1-70B-Instruct | Used to annotate randomly sampled code data for fine-tuning a small language model (i.e., Qwen2.5-0.5B). |
| Synthetic QA generation | Qwen2.5-72B-Instruct / Llama-3.3-70B-Instruct | Used to extract and refine question-answer pairs from web documents. |
| Code Translation | Qwen2.5-Coder-32B-Instruct / Llama-3.1-70B-Instruct | Used to translate code from other programming languages into Python. |
| Text & code block synthesis | Llama-3.1-70B-Instruct | Used to synthesize text and code blocks. |

# E   Summary of LLM Usage During Data Curation

# F   Training Details

## F.1   TinyLlama Training

In all ablation experiments, we keep our training hyper-parameter the same except for training steps. We present our full training details in Table 13.

Table 13: Training hyper-parameters.

| Hyper-parameter | 5B / 15B / 55B Tokens |
|---|---|
| Context Length | 2,048 |
| Batch Size | 1,024 |
| Max Steps | 2,500 / 7,500 / 27,500 |
| Warmup Steps | 0 |
| Weight Decay | 0.1 |
| Optimizer | AdamW |
| LR Scheduler | cosine |
| Learning Rate (LR) | 8e-5 → 8e-6 |

Table 14: Training Data Mixture for Llama-3.

| Data | Ratio % |
|---|---|
| DCLM | 10 |
| Web | 15 |
| Web-pro | 35 |
| Code | 2.5 |
| QA | 10 |
| Trans. code | 2.5 |
| Text & code block | 25 |
| **Total** | 100 |

Table 15: Training hyper-parameters.

| Hyper-parameter | Llama-3.2-1B / 3B |
|---|---|
| Context Length | 8,192 |
| Batch Size | 512 |
| Max Steps | 25,000 / 25,000 (stop at 12,500) |
| Warmup Steps | 0 |
| Weight Decay | 0.1 |
| Optimizer | AdamW |
| LR Scheduler | cosine |
| Learning Rate (LR) | 5e-5 → 5e-6
3e-5 → 3e-6 |

## F.2   Llama-3 Training

The data mixture and hyper-parameters for Llama-3 training are presented in Table 14 and Table 15.

# G  Evaluation Details and Full Results

## G.1  Full Benchmarks

Our core set of tasks and eval settings are at below. We revised our evaluation from DeepSeekMath (Shao et al., 2024): we fixed one of its prompts [3], and support PAL for more benchmarks.

1. GSM8K (Cobbe et al., 2021), 8-shot
2. MATH (Hendrycks et al., 2021), 4-shot
3. ASDiv (Miao et al., 2020), 8-shot
4. SVAMP (Patel et al., 2021), 8-shot
5. MAWPS (Koncel-Kedziorski et al., 2016), 8-shot

Our extended set of tasks are:

1. MMLU-STEM (Hendrycks et al., 2020), 4-shot
2. TabMWP (Lu et al., 2023), 8-shot
3. MathQA (Amini et al., 2019), 8-shot
4. SAT (Azerbayev et al., 2023), 4-shot
5. OCW Courses (Lewkowycz et al., 2022), 4-shot

## G.2  Full Ablation Results

We present our full results in this section:

1. For ablation on Web Data We provide the full ablation results on math text extraction, Minhash deduplication and `fastText` in Table 16, Table 17, Table 18.
2. For ablations on Code filtering, please see Table 19, and Table 20.
3. For ablations on synthetic data, please see Table 21, and Table 22.
4. The full comparison results are provided in Table 23.
5. For evaluation results for Llama-3, please see Table 24.

Table 16: Full ablation results on math text extraction within 15B-token training budget

| Text Extractors | w/ HTML Optimization | ASDiV | GSM8K | MATH | MATH-SAT | MATHQA | MAWPS | MMLU-STEM | OCW | SWAMP | TABMWP | Core Avg. | Ext. Avg. |
|---|---|---|---|---|---|---|---|---|---|---|---|---|---|
| TinyLlama-1.1B | - | 18.0 | 3.0 | 3.1 | 40.6 | 13.2 | 20.8 | 16.3 | 2.9 | 11.0 | 18.0 | 11.2 | 14.7 |
| trafilatura | ✗ | 32.6 | 5.9 | **4.3** | 21.9 | 12.9 | 44.8 | **23.2** | 2.2 | **22.3** | 21.8 | 22.0 | 19.2 |
| Resiliparse | ✔ | 33.5 | 5.8 | 3.9 | 15.6 | 10.9 | 47.3 | 21.3 | **2.6** | 22.1 | 22.7 | 22.5 | 18.6 |
| trafilatura | ✔ | **36.3** | **7.0** | 3.9 | **25.0** | **14.7** | **49.5** | 22.6 | 2.2 | 22.1 | **22.8** | **23.8** | **20.6** |

Table 17: Full ablation results on Minhash LSH within 55B-token training budget

| (r,b) | t | remaining tokens (B) | ASDiV | GSM8K | MATH | MATH-SAT | MATHQA | MAWPS | MMLU-STEM | OCW | SWAMP | TABMWP | Core. Avg. | Ext. Avg. |
|---|---|---|---|---|---|---|---|---|---|---|---|---|---|---|
| (14,9) | 0.70 | 16.0 | 26.1 | 4.9 | 3.4 | **25.0** | 10.0 | 36.0 | 20.5 | 2.6 | 16.3 | 21.0 | 17.3 | 16.6 |
| (14,8) | 0.75 | 23.5 | 29.1 | **5.4** | 3.7 | 17.5 | 9.3 | 38.6 | **23.1** | 1.5 | **18.7** | **23.1** | 19.1 | 17.0 |
| **(11,10)** | **0.75** | **26.0** | 29.8 | 4.4 | **3.9** | 23.1 | 10.2 | **41.4** | 19.3 | **2.9** | 17.6 | 22.1 | **19.4** | **17.5** |
| (11,11) | 0.75 | 25.0 | **30.1** | 4.3 | 3.8 | 9.4 | 10.9 | 38.9 | 21.0 | 2.1 | **18.7** | 20.3 | 19.2 | 16.0 |
| (9,12) | 0.80 | 29.0 | 28.3 | 4.4 | 3.6 | 18.8 | **11.2** | 40.3 | 21.8 | 2.4 | 17.5 | 20.7 | 18.8 | 16.9 |
| (9,13) | 0.80 | 30.0 | 27.7 | 3.5 | 3.5 | 13.8 | 10.0 | 36.7 | 21.7 | 2.1 | 16.6 | 21.1 | 17.6 | 15.7 |

---

[3] https://github.com/deepseek-ai/DeepSeek-Math/blob/main/evaluation/few_shot_prompts/pal_math_4_shot.py#L54

Table 18: Full ablation results on `fastText` within 5B-token training budget

| fastText version | ASDiV | GSM8K | MATH | MATH-SAT | MATHQA | MAWPS | MMLU-STEM | OCW | SWAMP | TABMWP | Core. Avg. | Ext. Avg. |
|---|---|---|---|---|---|---|---|---|---|---|---|---|
| V1: Open-Web-Math | 34.6 | 5.6 | 3.2 | 34.4 | 12.0 | 45.8 | 21.1 | 2.2 | 23.0 | 18.4 | 22.4 | 20.0 |
| V2: Random | 41.7 | 8.6 | 5.1 | 15.6 | 11.6 | 55.9 | 17.1 | 2.2 | 24.5 | 25.1 | 27.2 | 20.7 |
| V2: Balance | 41.3 | 8.9 | 5.0 | **28.1** | 15.5 | 57.8 | **19.2** | 2.2 | 26.2 | **26.2** | 27.8 | **23.0** |
| V2: Balance + CoT | **44.2** | **9.6** | **5.4** | 25.0 | **15.7** | **59.0** | 17.1 | 2.2 | **26.3** | 25.8 | **28.9** | **23.0** |

Table 19: Performance comparison of CoT and PAL under different filtering criteria

| Filter Criteria | CoT | | | | | |
|---|---|---|---|---|---|---|
| | GSM8K | MATH | ASDiV | MAWPS | SVAMP | Avg. |
| text only | 4.4 | 4.1 | 29.3 | 39.5 | 17.7 | 19.0 |
| $S\_$**edu** $\geq 3, S\_$**math** $\geq 3$ | 4.1 | 4.4 | 28.9 | 40.2 | 19.4 | 19.4 |
| $S\_$**edu** $\geq 3, S\_$**math** $\geq 4$ | 4.9 | 4.2 | 29.8 | 41.1 | 19.2 | 19.8 |
| $S\_$**edu** $\geq 4, S\_$**math** $\geq 3$ | 4.9 | 4.3 | 29.8 | 39.9 | 19.4 | 19.7 |
| $S\_$**edu** $\geq 4, S\_$**math** $\geq 4$ | 4.3 | 4.2 | 29.5 | 38.5 | 17.3 | 18.8 |

| Filter Criteria | PAL | | | | | |
|---|---|---|---|---|---|---|
| | GSM8K | MATH | ASDiV | MAWPS | SVAMP | Avg. |
| text only | 2.8 | 2.9 | 24.8 | 30.1 | 17.6 | 15.6 |
| $S\_$**edu** $\geq 3, S\_$**math** $\geq 3$ | 3.7 | 3.6 | 25.8 | 31.7 | 15.6 | 16.1 |
| $S\_$**edu** $\geq 3, S\_$**math** $\geq 4$ | 4.4 | 4.3 | 27.2 | 31.4 | 16.5 | 16.8 |
| $S\_$**edu** $\geq 4, S\_$**math** $\geq 3$ | 4.5 | 3.7 | 27.4 | 32.3 | 19.5 | 17.5 |
| $S\_$**edu** $\geq 4, S\_$**math** $\geq 4$ | 5.7 | 5.5 | 29.7 | 36.4 | 20.2 | 19.5 |

Table 20: Performance comparison of CoT and PAL under different mix ratios.

| Mix Ratio | CoT | | | | | |
|---|---|---|---|---|---|---|
| | GSM8K | MATH | ASDiV | MAWPS | SVAMP | Avg. |
| **text only** | 4.4 | 4.1 | 29.3 | 39.5 | 17.7 | 19.0 |
| **code:text = 1:7** | 3.8 | 4.2 | 30.2 | 40.0 | 18.3 | 19.3 |
| **code:text = 1:4** | 4.6 | 4.1 | 29.5 | 40.6 | 18.9 | 19.5 |
| **code:text = 1:2** | 3.9 | 4.0 | 28.3 | 27.6 | 18.1 | 16.4 |
| **code:text = 1:1** | 3.6 | 3.9 | 26.7 | 36.7 | 16.6 | 17.5 |

| Mix Ratio | PAL | | | | | |
|---|---|---|---|---|---|---|
| | GSM8K | MATH | ASDiV | MAWPS | SVAMP | Avg. |
| **text only** | 2.8 | 2.9 | 24.8 | 30.1 | 17.6 | 15.6 |
| **code:text = 1:7** | 4.4 | 4.3 | 27.2 | 31.8 | 19.5 | 17.4 |
| **code:text = 1:4** | 4.4 | 4.4 | 29.2 | 33.6 | 20.2 | 18.4 |
| **code:text = 1:2** | 4.3 | 4.4 | 27.6 | 34.1 | 17.1 | 17.5 |
| **code:text = 1:1** | 5.4 | 4.9 | 29.5 | 36.3 | 17.7 | 18.8 |

Table 21: Performance comparison of CoT using different Q&A datasets

| Data | ASDiV | GSM8K | MATH | MATH-SAT | MATHQA | MAWPS |
|---|---|---|---|---|---|---|
| **FM-4plus** | 41.9 | 10.5 | 6.1 | 34.4 | 14.4 | 57.9 |
| **WebInstruct** | 49.5 | 13.1 | 10.6 | 25.0 | 14.7 | 65.8 |
| **Vanilla Prompt** | 57.4 | 22.1 | 10.5 | 25.0 | 16.5 | 68.6 |
| **w. ELI5** | 58.6 | 25.9 | 12.3 | 21.9 | 18.4 | 71.6 |
| **w. ELI5 + IC** | 68.0 | 33.3 | 15.3 | 34.4 | 21.7 | 79.6 |

| Data | MMLU-STEM | OCW | SVAMP | TABMWP | Core Avg. | Ext. Avg. |
|---|---|---|---|---|---|---|
| **FM-4plus** | 20.6 | 2.9 | 25.3 | 25.5 | 28.3 | 19.6 |
| **WebInstruct** | 16.0 | 3.3 | 34.0 | 29.2 | 34.6 | 17.6 |
| **Vanilla Prompt** | 17.7 | 2.9 | 37.2 | 35.4 | 39.2 | 19.5 |
| **w. ELI5** | 15.6 | 4.0 | 38.1 | 35.9 | 41.3 | 19.2 |
| **w. ELI5 + IC** | 18.3 | 3.3 | 48.0 | 40.1 | 48.8 | 23.6 |

Table 22: Performance comparison of CoT and PAL under different mix ratios.

| Data | CoT | | | | | |
|---|---|---|---|---|---|---|
| | GSM8K | MATH | ASDiV | MAWPS | SVAMP | Avg. |
| **code** | 4.3 | 4.2 | 29.5 | 38.5 | 17.3 | 18.8 |
| **trans. code** | 3.5 | 4.3 | 30.2 | 39.3 | 17.8 | 19.0 |
| **text & code block** | 6.7 | 5.2 | 34.0 | 45.4 | 21.2 | 22.5 |
| **text & code block** (full) | 12.4 | 7.9 | 43.8 | 58.6 | 31.2 | 30.8 |

| Data | PAL | | | | | |
|---|---|---|---|---|---|---|
| | GSM8K | MATH | ASDiV | MAWPS | SVAMP | Avg. |
| **code** | 5.7 | 5.5 | 29.7 | 36.4 | 20.2 | 19.5 |
| **trans. code** | 7.0 | 5.3 | 31.3 | 39.4 | 20.1 | 20.6 |
| **text & code block** | 9.6 | 10.6 | 41.1 | 51.2 | 27.8 | 28.1 |
| **text & code block** (full) | 26.9 | 17.3 | 62.1 | 78.0 | 48.3 | 46.5 |

Table 23: Full comparison CoT results with existing corpora within 55B-token training budget

| Corpus | ASDiV | GSM8K | MATH | MATH-SAT | MATHQA | MAWPS | MMLU-STEM | OCW | SWAMP | TABMWP | Core. Avg. | Ext. Avg. |
|---|---|---|---|---|---|---|---|---|---|---|---|---|
| **MegaMath-Web-Pro (15B, Ours)** | **61.9** | **24.1** | **12.0** | 34.4 | 15.4 | **75.7** | **28.2** | 2.6 | **42.9** | 32.5 | **43.3** | **33.0** |
| FineMath-4+ (11B) | 55.7 | 21.1 | 11.4 | 31.3 | **23.7** | 70.9 | 25.9 | 2.6 | 35.9 | 32.6 | 39.0 | 31.1 |
| **MegaMath-Web-Top 50% (Ours)** | 53.0 | 15.5 | 8.4 | 31.3 | 15.5 | 68.3 | 25.7 | 3.7 | 33.1 | **32.7** | 35.6 | 28.7 |
| FineMath-3+ (41.6B) | 50.8 | 17.1 | 8.5 | 21.9 | 17.5 | 68.4 | 24.4 | **4.4** | 30.7 | 30.5 | 35.1 | 27.4 |
| **MegaMath-Web-Top 75% (Ours)** | 46.8 | 11.7 | 6.9 | **43.8** | 17.3 | 62.4 | 17.2 | 2.6 | 29.2 | 30.0 | 31.4 | 26.8 |
| InfiMM-WebMath (55B) | 46.1 | 12.3 | 6.4 | 25.0 | 15.3 | 63.0 | 22.5 | 3.3 | 26.3 | 28.4 | 30.8 | 24.9 |
| **MegaMath-Web-Full (Ours)** | 44.7 | 11.6 | 6.4 | 25.0 | 13.0 | 61.0 | 21.8 | 2.2 | 26.2 | 30.0 | 30.0 | 24.2 |
| Open-Web-Math (14.5B) | 39.7 | 8.7 | 6.3 | 31.3 | 12.9 | 54.2 | 22.7 | 2.6 | 24.1 | 25.1 | 26.6 | 22.8 |

Table 24: Full results of training MegaMath on Llama-3 series of models.

| Model | CoT | | | | | |
|---|---|---|---|---|---|---|
| | ASDiV | GSM8K | MATH | MAWPS | SVAMP | Avg. |
| **Llama-3.2-1B** | 33.8 | 8.5 | 4.6 | 43.3 | 21.5 | 22.3 |
| **MegaMath Llama-3.2-1B** | 59.8 | 25.7 | 9.5 | 74.6 | 41.3 | 42.2 |
| **Llama-3.2-3B** | 60.5 | 30.1 | 9.2 | 80.5 | 52.6 | 46.6 |
| **MegaMath Llama-3.2-3B** | 78.8 | 56.2 | 25.1 | 90.2 | 71.6 | 64.4 |

| Model | PAL | | | | | |
|---|---|---|---|---|---|---|
| | ASDiV | GSM8K | MATH | MAWPS | SVAMP | Avg. |
| **Llama-3.2-1B** | 13.4 | 7.9 | 3.1 | 16.8 | 8.4 | 9.9 |
| **MegaMath Llama-3.2-1B** | 42.8 | 16.8 | 6.3 | 52.7 | 29.8 | 29.7 |
| **Llama-3.2-3B** | 65.1 | 35.7 | 0.4 | 83.3 | 58.3 | 48.6 |
| **MegaMath Llama-3.2-3B** | 78.1 | 55.7 | 24.6 | 93.7 | 74.4 | 65.3 |

