# OpenReview forum: "MegaMath: Pushing the Limits of Open Math Corpora"
_colmweb.org/COLM/2025/Conference — COLM 2025_

### Official Review · Reviewer_PQwQ · 2025-05-08

**Rating:** 8
**Confidence:** 4
**Ethics Flag:** 1

**Summary:**

This paper presents MegaMath, a carefully curated math corpus for pre-training LLMs.
The construction pipeline is generally valid: (i) HTML extraction with fastText filtering & deduplication (ii) Math-related code data recall from Stack-v2 (iii) Data synthesis to create QA style data, translated Python code, and text-code blocks.
The final MegaMath comprises 371B tokens, the largest open math pretraining corpus.
The curation pipeline and hyperparameter settings are validated using TinyLLaMA-1B and continual pre-training with LLaMA 3.2 justifies the value of the MegaMath.

**Questions To Authors:**

## Questions
- I am curious about the utility of MegaMath-Synthetic, but did not find any experiments to investigate this. Sec 3.4 is operated on data synthesized from Open-Web-Math and Infimm-Web-Math which may not transfer to the MegaMath?

- There are various models adopted for scoring, annotation, and synthesis. It would be better to aggregate the LLMs choices into ONE table for better clarity. For example, Line 288 states `with LLM-annotated math-related document` and Line 316 ` using the latest and most capable LLMs.`  without specifying the used LLM.




## Typos

Line337:  Also, These -> Also, these

Caption of Figure 4: ; Right: text & ode block data curation -> ; Right: text & code block data curation

**Reasons To Accept:**

- The curated MegaMath is of great value for LLM pre-training and could benefit future studies towards better reasoning models.
- The overall pipeline is sound and well-executed comprehensive validation experiments under academic resources.

**Reasons To Reject:**

I did not find a strong reason to reject this paper.

---

> ### Author Response · Authors · 2025-06-02
> **Response to Reviewer PQwQ [2/2]**
>
> $\textrm{\color{blue}Question 2}$
> ```
> There are various models adopted for scoring, annotation, and synthesis. It would be better to aggregate the LLMs choices into ONE table for better clarity. For example, Line 288 states with LLM-annotated math-related document and Line 316 using the latest and most capable LLMs. without specifying the used LLM.
> ```
>
> Thank you for your suggestion! We agree that aggregating the LLM choices into a single table would improve clarity. Below, we provide a consolidated summary of all LLMs used throughout the pipeline for scoring, annotation, and synthesis. We will also include this table in the revised version of the submission to enhance transparency and reproducibility.
>
> | **Purpose / Stage**                      | **LLM Used**                                         | **Description / Notes**                                                                                  |
> |------------------------------------------|------------------------------------------------------|----------------------------------------------------------------------------------------------------------|
> | Math-related web document classification |  Llama-3.1-70B-Instruct                              | Used to annotated seed documents for training FastText classifiers (Line 116, 288)                       |
> | Web document refinement                  |  Llama-3.3-70B-instruct                              | Used to further remove noise, refine the web text into higher quality, similar to Nemontron-CC.          |
> | Math-related code classification         |  Llama-3.1-70B-Instruct                              | Used to annotate  randomly sampled code data for small language model (i.e., Qwen-2.5-0.5B) fine-tuning. |
> | Synthetic QA generation                  | Qwen-2.5-72B-Instruct /  Llama-3.3-70B-Instruct      | Used to extract and refine QA from web documents. (Line 316)                                             |
> | Code Translation                         | Qwen2.5-Coder-32B-Instruct /  Llama-3.1-70B-Instruct | Used to translate code from other programming languages into Python.                                     |
> | Text & code block synthesis              |  Llama-3.1-70B-instruct                              | Used for synthesize text & code block.                                                                   |
>
> ---
>
> $\textrm{\color{blue}About Typos}$
>
> Thank you for your careful reading! We will correct these typos in the revised version and perform another thorough round of proofreading.

---

> > ### Comment · Reviewer_PQwQ · 2025-06-04
> >
> > Thank you for your response and updated results, which effectively address my concerns.
> >
> > I have no further concerns, and I believe this paper warrants acceptance due to its strong contributions to the open LLM community.

---

> > > ### Author Response · Authors · 2025-06-05
> > >
> > > Thank you for your thoughtful feedback and kind support of MegaMath! We truly appreciate your recognition of our work and are delighted that our responses have successfully addressed your concerns. Your encouragement means a great deal to us, and we’re excited to contribute to the open LLM community with this paper.

---

> ### Author Response · Authors · 2025-06-02
> **Response to Reviewer PQwQ [1/2]**
>
> We sincerely thank the reviewer for the encouraging and constructive feedback. We appreciate the recognition of our corpus’s practical value, the systematic ablation studies, and the empirical effectiveness of our approach. For your questions:
>
> $\textrm{\color{blue}Question 1}$
> ```
> I am curious about the utility of MegaMath-Synthetic, but did not find any experiments to investigate this. Sec 3.4 is operated on data synthesized from Open-Web-Math and Infimm-Web-Math which may not transfer to the MegaMath?
> ```
>
> Thank you for highlighting this point — we acknowledge that we did not explicitly present an evaluation of the final MegaMath-Synthetic dataset in the original submission. We would like to clarify that the data synthesis pipeline described in Section 3.4 (Tables 6 and 7) was developed during the early stage of our project, when the MegaMath dataset was still under construction. To build and validate the pipeline efficiently, we initially used existing open-source web math datasets (Open-Web-Math and Infimm-Web-Math), which offered sufficient coverage and structure for pipeline development.
>
> We have now supplemented the same experiments used in Section 3.4 (i.e., Table 6 and Table 7) with results based on the final MegaMath-Synthetic data.
> For Table 6, we have added the evaluation results using the final MegaMath-Synth-QA, and the results are as follows:
>
> | **Data**                              | **Core Avg.** | **Ext. Avg.** |
> |---------------------------------------|---------------|---------------|
> | FM-4plus                              | 28.3          | 19.6          |
> | WebInstruct                           | 34.6          | 17.6          |
> | Vanilla Prompt                        | 39.2          | 19.5          |
> | w. ELI5                               | 41.3          | 19.2          |
> | w. ELI5 + IC                          | 48.8          | 23.6          |
> | **MegaMath-Synth-QA (final version)** | **49.6**      | **23.8**      |
>
> For Table 7, the code and translated code rows already correspond to the final MegaMath dataset; for the text & code block row, we also report the results based on the final version of this subset:
>
> | **Data**                                    | **CoT Avg.** | **PAL Avg.** |
> |---------------------------------------------|--------------|--------------|
> | code                                        | 18.8         | 19.5         |
> | trans. code                                 | 19.0         | 20.6         |
> | text & code block                           | 22.5         | 28.1         |
> | text & code block (full)                    | 30.8         | 46.5         |
> | **text & code block(full & final version)** | **33.5**     | **46.3**     |

---

### Official Review · Reviewer_CT6F · 2025-05-11

**Rating:** 6
**Confidence:** 4
**Ethics Flag:** 1

**Summary:**

This paper proposes a systematic methodology for developing a large-scale mathematical pretraining corpus. MegaMath comprises three main types of pretraining data:

- First, the Web math pretraining data optimizes HTML extraction techniques to preserve complete mathematical expressions and conducts filtering and deduplication, providing valuable experimental insights. The authors further refine this data by using classifiers and large language models to extract higher-quality subsets.
- Second, the Code data employs small language models to recall mathematics-related code and offers empirical guidance on optimal mixing ratios for pretraining.
- Third, the Synthetic data includes question-answer pairs extracted from raw documents and refined using LLMs, translations of code from other languages into Python, and "generated mathematical code" consisting of interleaved text, symbolic expressions, and code blocks.

The resulting pretraining dataset reaches industrial scale at 371B tokens. The authors conduct comprehensive ablation experiments to validate key design choices and demonstrate considerable performance improvements when training the LLaMA-3 series models with their corpus.

**Reasons To Accept:**

- This work provides an industrial-scale mathematical pretraining corpus spanning web data, code mixtures, and synthetic data. To my knowledge, this approach closely resembles industrial practices and offers significant value to the research community by providing extensive empirical references and practical guidelines.

- The ablation studies validating each key design choice are comprehensive and methodical. The paper systematically compares different text processing techniques (trafilatura vs. Resiliparse), math-optimized HTML processing methods, and deduplication strategies with various parameters, providing valuable insights for domain-specific dataset construction.

- The performance improvements on the LLaMA-3 series models are substantial and demonstrate the effectiveness of the proposed methods and the quality of the constructed corpus.

**Reasons To Reject:**

- The evaluation results show puzzling discrepancies with LLama-3.2's official results. For example, Llama-3.2-1B using CoT 4-shot on MATH achieves only 4.6 in this paper, whereas the official report states that Llama-3.2-1B on MATH (CoT) 0-shot reaches 30.6. Similarly, Llama-3.2-1B using CoT on GSM8K scores only 8.5 in this paper, while officially Llama-3.2-1B on GSM8K (CoT) 8-shot achieves 44.4. These discrepancies require explanation.

- All experiments are conducted solely on LLaMA-3.2, leaving uncertainty about whether the method would yield similar improvements on other advanced open-source pretrained models such as Qwen2.5-Base. This lack of cross-model validation weakens the generalizability claims of the approach.

- For the final training phase, the authors select 100B tokens from the total 371B corpus, but provide insufficient details on the selection criteria. What is the mixture configuration of this data? Did the authors conduct further experiments on different mixture configurations? This information is crucial for understanding the method's practicality and reproducibility.

- Regarding the synthetic data, particularly the QA pairs, the paper lacks thorough analysis comparing their quality to human-written data. Data generated using LLMs may introduce model biases or inaccuracies, and this risk has not been systematically evaluated.

- While the paper introduces synthetic QA pairs, it does not thoroughly explore their implications, such as whether they might exhaust the model's potential. How would the model perform if subsequently fine-tuned on QA pairs in SFT stage? This analysis would be valuable for understanding the interaction between synthetic data in pretraining and fine-tuning stages.

- The paper primarily focuses on model performance on benchmarks like GSM8K and MATH, but lacks evaluation on more complex mathematical problems such as advanced competition questions (e.g., AIME, Omni-Math, CollegeMath, OlympiadBench). This limitation makes it difficult to assess MegaMath's effectiveness in developing deep mathematical reasoning capabilities, particularly for more challenging mathematical problems.

---

> ### Author Response · Authors · 2025-06-02
> **Response to Reviewer CT6F [5/5]**
>
> $\textrm{\color{blue}Question 6}$
> ```
> The paper primarily focuses on model performance on benchmarks like GSM8K and MATH, but lacks evaluation on more complex mathematical problems such as advanced competition questions (e.g., AIME, Omni-Math, CollegeMath, OlympiadBench). This limitation makes it difficult to assess MegaMath's effectiveness in developing deep mathematical reasoning capabilities, particularly for more challenging mathematical problems.
> ```
>
> We appreciate your insightful comment. While our primary evaluation focuses on standard mathematical reasoning benchmarks like GSM8K and MATH, we did include some more challenging datasets, such as OCW (as used in DeepSeekMath), and we have also conducted additional evaluations on benchmarks like OlympiadBench, and AMC at below.
>
> | **Model**                 | **OlympiadBench** | **AMC**   |
> |---------------------------|-------------------|-----------|
> | Llama-3.2-1B              | 1.04              | 0.0       |
> | **MegaMath-Llama-3.2-1B** | **2.67**         | **7.50** |
> | Llama-3.2-3B              | 2.37              | 5.0       |
> | **MegaMath-Llama-3.2-3B** | **5.48**          | **7.50**  |
>
> We also would like to clarify that extremely difficult benchmarks (e.g., AIME or Olympiad-level tasks) often do not reflect improvements in pretraining alone. For instance, Qwen-2.5-Math, which was extensively trained on over 1T tokens and achieved state-of-the-art performance on standard math benchmarks at the time, still performs poorly on AIME-level questions. This phenomenon is not unique to Qwen-2.5; even the most capable proprietary models perform weakly on these tasks:
>
>  **>>> the results below are from qwen-2.5 math technical report**
> | **Model**      | **AIME24 Score** |
> |----------------|------------------|
> | Claude 3 Opus  | 2 / 30           |
> | GPT-4 Turbo    | 1 / 30           |
> | Gemini 1.5 Pro | 2 / 30           |
>
> This suggests that solving high-level competition problems is likely beyond the reach of pretraining alone. Such tasks typically require targeted post-training strategies, including carefully designed SFT datasets and reinforcement learning-based optimization.
>
> **Meanwhile, our MegaMath is presented as a pretraining dataset aimed at enhancing foundational mathematical reasoning abilities. Our goal is not to directly compete on the most difficult benchmarks, which would require significant additional fine-tuning and computational resources far beyond the current scope. We believe this is a difference in focus rather than a limitation of MegaMath’s effectiveness.**
>
> In summary, MegaMath aims to improve the general reasoning capabilities of base models through high-quality pretraining data. Tackling advanced competition-level problems is an important future direction, but we believe it requires dedicated post-training rather than pretraining alone.

---

> ### Author Response · Authors · 2025-06-02
> **Response to Reviewer CT6F [4/5]**
>
> $\textrm{\color{blue}Question 4}$
> ```
> While the paper introduces synthetic QA pairs, it does not thoroughly explore their implications, such as whether they might exhaust the model's potential. How would the model perform if subsequently fine-tuned on QA pairs in SFT stage? This analysis would be valuable for understanding the interaction between synthetic data in pretraining and fine-tuning stages.
> ```
>
> Thank you for the thoughtful question. We’d like to clarify that the synthetic QA pairs were designed for pretraining purposes. While they exhibit a QA structure, they do not follow standard instruction formats, and many examples interweave question and answer in a single block—making them less suited for direct use in SFT. (i.e., applying loss only to the response segment). As such, these data may not be directly suitable for SFT.
>
> Your point about the interaction between synthetic pretraining data and downstream SFT is also a very important one. Prior work such as MiniCPM [2] (i.e., Table 1 in MiniCPM Paper) has shown that incorporating high-quality, instruction-style QA data during the later stages of pretraining (e.g., via decayed learning rate) can lead to better downstream performance after SFT. This suggests that exposing models to such data in pretraining does not exhaust the model’s capacity, but rather can enhance the effect of subsequent fine-tuning.
> In our case, although synthetic QA data are not formatted for SFT, **they likely serve a similar role: exposing the model to multi-step reasoning patterns and answer-focused objectives during pretraining. Exploring how fine-tuning on clean QA data interacts with such synthetic pretraining is indeed a valuable direction, and we agree that further empirical analysis here would strengthen our understanding**. We plan to include this in future work.
>
> [2] MiniCPM: Unveiling the Potential of Small Language Models with Scalable Training Strategies
>
>
> $\textrm{\color{blue}Question 5}$
> ```
> For the final training phase, the authors select 100B tokens from the total 371B corpus, but provide insufficient details on the selection criteria. What is the mixture configuration of this data? Did the authors conduct further experiments on different mixture configurations? This information is crucial for understanding the method's practicality and reproducibility.
> ```
> Thank you for your thoughtful question.
>
> In fact, we have provided the detailed mixture configuration of the 100B-token subset used in the final training phase in Appendix $\textrm{\color{blue}Table 12}$. For clarity, we also reproduce it below:
>
> | **Category**      | **Ratio (%)** |
> |-------------------|---------------|
> | DCLM              | 10            |
> | Web               | 15            |
> | Web-pro           | 35            |
> | Code              | 2.5           |
> | QA                | 10            |
> | Trans. code       | 2.5           |
> | Text & code block | 25            |
> | **Total**         | **100**       |
>
> As for the mixture design principle, we followed general practices from previous works on math-focused continual pretraining, such as DeepSeek-Math [3] and Llemma [4]. Specifically, we adopted a high-level ratio of:
> - Web (DCLM) : Math-related text (Web, Web-Pro, QA) : Code (Code, Trans. Code, Text & Code Block) = 10 : 60 : 30
>
> We did not conduct extensive tuning over different mixture variants. Instead, our guiding principle was to include a representative mix of all high-quality data sources from our corpus in a balanced way. For example, we gave more weight to `Web-Pro` and `Text & Code Block`, as they were found to have higher signal quality in our filtering stages. At the same time, we aimed to minimize redundancy and domain over-representation. This design reflects our objective to showcase the combined strength and diversity of the MegaMath dataset in a practical training setup.
>
> [3] DeepSeekMath: Pushing the Limits of Mathematical Reasoning in Open Language Models.
>
> [4] Llemma: An Open Language Model For Mathematics.

---

> ### Author Response · Authors · 2025-06-02
> **Response to Reviewer CT6F [3/5]**
>
> $\textrm{\color{blue}Question 3}$
> ```
> Regarding the synthetic data, particularly the QA pairs, the paper lacks thorough analysis comparing their quality to human-written data. Data generated using LLMs may introduce model biases or inaccuracies, and this risk has not been systematically evaluated.
> ```
> Thank you for raising this important point.
>
> We also agree that synthetic data—especially when generated by LLMs—may introduce biases or inaccuracies, and its quality must be carefully assessed. While we did not conduct a direct comparison with human-authored QA pairs in this version of the paper, we included several indirect evaluations in Section 3.4 to assess the utility and robustness of our synthetic data:
> - $\textrm{\color{blue}Table 6}$ shows that QA data generated via our carefully designed prompting strategy (emphasizing information completeness and mathematical precision) significantly improves downstream performance over naive generation baselines.
> - $\textrm{\color{blue}Table 7}$ further demonstrates that synthetic subsets—such as translated code and text & code blocks—consistently enhance model performance, suggesting that the generated content is both useful and reliable.
>
> Also, to improve data quality and relavance to math-centric tasks, we applied a multi-stage pipeline such as:
> - Stringent filtering via training robust and accurate classifiers
> - Deduplication to reduce redundancy
> - AST-based validation for code integrity
>
> We also want to highlight that our entire data generation and filtering pipeline is highly reproducible, including:
> - Prompt templates and generation scripts
> - All filtering and validation mechanisms
> - The released datasets themselves
>
> We believe this level of transparency not only supports reproducibility but also enables the community to audit, adapt, or improve upon our methods. Furthermore, we recognize that a systematic, human-grounded comparison between synthetic and human-authored datasets is a valuable direction, and we plan to pursue this in future work using both expert evaluation and more fine-grained analysis.

---

> ### Author Response · Authors · 2025-06-02
> **Response to Reviewer CT6F [2/5]**
>
> $\textrm{\color{blue}Question 2}$
> ```
> All experiments are conducted solely on LLaMA-3.2, leaving uncertainty about whether the method would yield similar improvements on other advanced open-source pretrained models such as Qwen2.5-Base. This lack of cross-model validation weakens the generalizability claims of the approach.
> ```
>
> Thank you for the valuable feedback. We would like to clarify a few key points.
>
> **1. Why we employ a proxy model**
>
> Building a large-scale pre-training dataset is inherently iterative. At each stage we need fast, informative feedback loops; a proxy model provides that by converting data-pipeline changes into measurable downstream performance signals without incurring the full cost of training a very large model. For a fair comparison, **the Table at below** summarizes how recent math-oriented corpora evaluate intermediate filtering strategies:
>
> | **Dataset**                | **Inter-stage Eval?**                              | **Inter-stage Model** | **Final Data Eval?** | **Final-stage Model** | **Metrics**                               |
> |----------------------------|----------------------------------------------------|-----------------------|----------------------|-----------------------|-------------------------------------------|
> | OpenWebMath                | No                                                 | –                     | Yes                  | Pythia-1.4B           | PPL / Accuracy on GSM / MATH / MultiArith |
> | InfiniMath-Web-Math (text) | No                                                 | –                     | Yes                  | DeepSeek-Coder-Base   | Accuracy on GSM / MMLU-STEM               |
> | FineMath                   | Yes (LM filtering)                                 | Llama-3.2–3B          | Yes                  | Llama-3.2–3B          | Accuracy on GSM / MATH                    |
> | **MegaMath (ours)**        | **Yes (FastText, dedup, LM filtering, synthesis)** | **TinyLlama-1.1B**    | **Yes**              | **Llama-3.2–1B / 3B** | **Accuracy on 10 math benchmarks**        |
>
> Our goal is to let intuitive downstream scores guide data quality. Hence Section 3.3 to Section 3.5 relies heavily on a proper model to traverse the full pipeline, monitor gains, and refine individual filtering steps. **To the best of our knowledge, we are the only work that conducts model-based evaluations with accuracy metrics at nearly every stage of dataset construction.**
>
> **2. Choice of proxy model: TinyLlama-1.1 B, not Llama-3.2**
>
> you mentioned that:
> ```
> All experiments are conducted solely on LLaMA-3.2
> ```
>
> All ablation studies (Sec. 3.2-3.5) use TinyLlama-1.1 B—a clean, purely “base” model trained for 3T tokens with a simple cosine schedule and without any SFT or post-training mixture. Its frugal size and clean recipe make it especially sensitive to data-quality changes, which is exactly what we need during early iterations.
> Conversely, we avoided Llama-3.2 as a proxy. Because its undisclosed training schedule blends annealed data and high-quality SFT tokens, it can mask subtle improvements (or degradations) introduced by our pipeline.
>
> **3. Role of Llama-3.2 in the paper**
>
> Only after the dataset was finalized did we fine-tune a powerful model—Llama-3.2-1B/3B—to showcase MegaMath’s full potential across ten public math benchmarks (Sec. 3.6). This step is meant as a validation of the finished corpus, not as part of the proxy-based iteration loop.
>
> **4. Focus of the paper**
>
> Finally, we would like to emphasize that our contribution is not about continual pre-training over an existing base model. Instead, our primary goal is to present a transparent and replicable blueprint for constructing a large-scale mathematical pre-training dataset from scratch. In this process, lightweight proxy models are an essential engineering tool to enable fast iteration and efficient pipeline optimization. Stronger models, in contrast, are used at the end to validate the final dataset and demonstrate its impact.
>
> We fully agree that validating MegaMath with more models would also be interesting. For example, Qwen-2.5-Base already performs strongly on math tasks, benefiting from advanced training strategies such as data decay and annealing. Evaluating MegaMath on such models would offer a compelling signal of its quality. However, we note that to fully unlock the potential of powerful models like Qwen-2.5, high-quality QA-style reasoning data (e.g., NuminaMath-style competition-level QA data) is likely essential. In this context, performance gains may depend more on supplementing MegaMath with curated reasoning data than on pre-training with MegaMath alone—a promising direction for future work.

---

> ### Author Response · Authors · 2025-06-02
> **Response to Reviewer CT6F [1/5]**
>
> We sincerely thank the reviewer for the thoughtful and encouraging feedback! We are especially grateful for the recognition of our efforts in constructing an industrial-scale mathematical pretraining corpus and for highlighting the practical value it brings to the research community. Your comments are highly motivating and affirm the contributions we hoped to make with this work.
>
> We are also happy to address all your questions and clarify the implementation details.
>
> $\textrm{\color{blue}Question 1}$
> ```
> The evaluation results show puzzling discrepancies with LLama-3.2's official results. For example, Llama-3.2-1B using CoT 4-shot on MATH achieves only 4.6 in this paper, whereas the official report states that Llama-3.2-1B on MATH (CoT) 0-shot reaches 30.6. Similarly, Llama-3.2-1B using CoT on GSM8K scores only 8.5 in this paper, while officially Llama-3.2-1B on GSM8K (CoT) 8-shot achieves 44.4. These discrepancies require explanation.
> ```
>
> We appreciate your careful comparison of evaluation results and are happy to clarify the discrepancy.
> However, after a thorough investigation, we believe this may be due to a misunderstanding regarding the source and type of models used. The results you referenced (e.g., **MATH: 30.6, GSM8K: 44.4**) appear to come from Meta's official blog post [1], **which reports scores from instruction-tuned models**—**specifically meta-llama/Llama-3.2-1B-Instruct**—evaluated on what they describe as a "lightweight instruction-tuned benchmark."
>
> In contrast, **our reported scores are based on the base models** (e.g., meta-llama/Llama-3.2-1B), without any instruction tuning or SFT applied. All evaluations in our paper are performed on base models, as clearly indicated in $\textrm{\color{blue}Figure 6}$ and $\textrm{\color{blue}Table 22}$. This distinction is critical, as instruction tuning significantly boosts performance on CoT-style tasks such as MATH and GSM8K; while in MegaMath, we primarily focus only on base model performance.
>
> We hope this clears up your confusion.
>
> [1] Meta AI Blog — Introducing Llama 3.2: https://ai.meta.com/blog/llama-3-2-connect-2024-vision-edge-mobile-devices/

---

> ### Comment · Reviewer_CT6F · 2025-06-05
>
> Thank you for the comprehensive responses that address my concerns.
>
> Given your emphasis that "the primary goal of our paper is to transparently demonstrate how to construct a large-scale mathematical pre-training dataset from scratch," would you consider open-sourcing the complete data construction pipeline code and models? This would enable the LLM community to truly leverage your transparent and scalable framework for constructing mathematical pre-training datasets. The value of open-sourcing the engineering code corresponding to your methodology would be great for the community.

---

> > ### Author Response · Authors · 2025-06-05
> >
> > Thank you for your quick response!  We’re glad to hear that our clarifications addressed your concerns. We strongly believe in the value of open-source contributions and are committed to fostering transparency, reproducibility, and collaboration in the community. To that end, we will release the complete codebase and models used in constructing our large-scale mathematical pre-training dataset. Specifically, we will open-source:
> >
> > - **the full Web data pipeline**, including all scripts for data downloading scripts, URL filtering, language identification, the two-stage text extraction process and the two-stage fasttext filtering module (along with our trained models), and our deduplication pipeline.
> >
> > - **the code data pipeline**, especially the classifier we trained to filter code with mathematical relevance.
> >
> > - **the synthetic data generation pipeline**, including QA extraction and refinement processes, code translation components, and our improved pipeline for generating text & code blocks.
> >
> > We hope this end-to-end transparency not only enables others to reproduce our results but also provides a solid foundation for future work in mathematical data construction at scale.

---

> > > ### Comment · Reviewer_CT6F · 2025-06-08
> > >
> > > Thank you for your commitment to fully open-sourcing the code and models. This complete transparency will benefit the community, whereas partial release would have much less impact.
> > >
> > > I intend to maintain my current score, which is in favor of the acceptance of this paper.

---

### Official Review · Reviewer_J7PA · 2025-05-13

**Rating:** 6
**Confidence:** 3
**Ethics Flag:** 1

**Summary:**

This paper introduces MegaMath, a large-scale, high-quality corpus targeting LLMs' mathematical reasoning capabilities. The corpus has about 371.6B tokens derived from web data (279B tokens), code (28.1B tokens), and synthetic data (64.5B tokens). The paper details the curation process across three main components: (1) Re-extracting mathematical documents from Common Crawl with math-oriented HTML optimizations, (2) Filtering high-quality math-related code from existing code corpora, and (3) Synthesizing QA-style text, math-related code, and code-interpreter style data. The authors validate their approach through extensive ablation studies and demonstrate significant improvements when training Llama-3.2 models on their dataset, achieving substantial gains in mathematical reasoning benchmarks.

**Questions To Authors:**

- Qwen models add the SFT dataset to pretraining/mid-training. Have you considered adding NuminaMath/OpenMathInstruct to pretraining to supplement the synthesized data in Section 3.4?
- What do you mean by "information completeness" in Section 3.4?
- How does improving math reasoning capabilities in pretraining impact other tasks?
- How do the different subsets of the dataset - web, code, and synthesized data?

**Reasons To Accept:**

- The proposed resource, i.e., MegaMath, would have a significant benefit to the open-source community.
- Most of the dataset design choices are backed by ablation studies.
- The empirical results clearly demonstrate the edge of MegaMath over existing alternatives.
- The work seems highly reproducible, with detailed methodologies that would allow others to build upon this research.

**Reasons To Reject:**

- The paper does a thorough job of execution but lacks novelty.
- The impact of pretraining for math reasoning is not clear after post-training. Would the gains remain post-alignment as well?
- Some of the key details are not well explained. What is the CoT data that the authors are talking about? What is its source? Similarly "information completeness" is introduced without any context in Section 3.4.

---

> ### Author Response · Authors · 2025-06-03
> **Response to Reviewer J7PA [3/3]**
>
> $\textrm{\color{blue}Question 5}$
> ```
> What do you mean by "information completeness" in Section 3.4?
> ```
> Thank you for raising this point. We will provide a more detailed clarification in the revised version of the paper.
>
> To elaborate here:
>
> although we include the full prompt in the Appendix, the term “information completeness” actually refers to the requirement that each extracted QA pair be self-contained and mathematically solvable without needing to refer back to the source document.
> This issue arose because prior works (e.g., MAmmoTH2[4]) either did not release their prompts or provided insufficient detail about the document structure and content. **As a result, in MegaMath, we built our own QA extraction pipeline from scratch and shared our findings.**
>
> We observed that, without proper prompting, the model sometimes generated QA pairs that were dependent on document context—for example, questions like “According to the document, how many more apples does Tom have than Sam?” Such questions cannot be understood or answered independently and therefore lack information completeness.
>
> To address this, we included an explicit instruction in our prompt:
> ```
> The extracted pair must be self-contained and mathematically precise, allowing independent solving without additional context.
> ```
>  This adjustment significantly improved the quality of the generated data: it increased the average token count per QA pair by approximately 20%, and led to better data quality, as shown in $\textrm{\color{blue}Table 6}$.
>
> [4] MAmmoTH2: Scaling Instructions from the Web
>
> $\textrm{\color{blue}Question 6}$
> ```
> How does improving math reasoning capabilities in pretraining impact other tasks?
> ```
> This is indeed a highly open-ended research question. In our evaluation setup, we intentionally divided the benchmark into core and extended sets to explore this aspect. Notably, the extended set includes tasks that are not strictly mathematical reasoning benchmarks—for example, MMLU-STEM and TabMWP. MMLU-STEM, in particular, contains questions from a variety of domains such as computer science, physics, and engineering.
>
> As shown in **Table 21**, models trained on MegaMath-Web-Pro achieved higher scores on MMLU-STEM compared to baselines, suggesting that pretraining on high-quality math data may lead to performance gains even in broader scientific and technical domains. Additionally, we observed that training with the text & code block subset improved the model’s ability to solve math problems using Python, which may also imply enhanced capabilities in code generation.
>
> Beyond our findings, several recent industrial works—such as the Qwen 2.5 Tech Report and DeepSeekMath—have also emphasized the importance of reasoning-focused data. Within the broader category of reasoning tasks, math and code are among the most frequently highlighted components. We believe this trend may indicate that math reasoning contributes to strengthening general reasoning abilities across diverse tasks.
>
> While this remains an open area of research, we hope these observations provide some insight into how improving math reasoning in pretraining can positively influence other downstream tasks.
>
> $\textrm{\color{blue}Question 7}$
> ```
> How do the different subsets of the dataset - web, code, and synthesized data?
> ```
> Thank you for your question. However, we found the sentence to be incomplete—“How do the different subsets of the dataset – web, code, and synthesized data?”—which makes it difficult for us to fully understand the intent behind it. We would greatly appreciate it if you could clarify or elaborate on the specific aspect you would like us to address. We are more than happy to provide a detailed response once we better understand your concern.

---

> ### Author Response · Authors · 2025-06-03
> **Response to Reviewer J7PA [2/3]**
>
> $\textrm{\color{blue}Question 3}$
> ```
> Some of the key details are not well explained. What is the CoT data that the authors are talking about? What is its source?
> ```
>
> We would be happy to clarify these points. However, it seems that you did not specify which section the CoT data in question refers to.
>
> **1. If you are referring to the CoT data used in the FastText classifier, we are happy to elaborate:**
> - About usage: We prepared 250K problems with well-structured, step-by-step solutions and combined them with high-scoring documents filtered by the LLM to form the positive seed dataset for FastText training, totaling 1 million examples. Meanwhile, we used 1 million low-scoring (score < 4) filtered documents as the negative seed dataset.
> - About the source:
>  We sampled 20K examples from [orca-math-word-problems](https://huggingface.co/datasets/microsoft/orca-math-word-problems-200k) and 230K from [OpenR1-Math](https://huggingface.co/datasets/open-r1/OpenR1-Math-220k)  to ensure broad coverage across different knowledge domains and difficulty levels, ranging from K–12 to competition-level problems. Both datasets feature detailed step-by-step solutions. Similar practice is also performed in DCLM-baselines[3], which incorporates instruction-formatted data such as OpenHermes 2.5 and well-explained documents from sources like the r/ExplainLikeImFive (ELI5) subreddit. In our case, we integrate problem-solving CoT data into the training process. We hypothesize that incorporating CoT data enhances the classifier’s ability to recall examples with intensive reasoning patterns—an effect supported by the downstream performance improvements shown in **Fig. 5**.
>
> [3] DataComp-LM: In search of the next generation of training sets for language models.
>
> **2. If you are instead referring to how CoT performance was evaluated, we primarily followed the evaluation framework from DeepSeekMath, where we assessed the model using few-shot CoT prompting.**
>
> $\textrm{\color{blue}Question 4}$
> ```
> Qwen models add the SFT dataset to pretraining/mid-training. Have you considered adding NuminaMath/OpenMathInstruct to pretraining to supplement the synthesized data in Section 3.4?
> ```
> This is a very interesting and open research question.
>
> **But first, we would like to clarify that our work does not focus on pre-training or continual pre-training over an existing base model.**
> **Instead, the primary goal of our paper is to transparently demonstrate how to construct a large-scale mathematical pre-training dataset from scratch.**
> Section 2 outlines our full data pipeline and design practices, while Section 3 evaluates the effectiveness of these practices through pre-training experiments. This is also the central focus of Section 3.4.
>
> Given this, incorporating datasets like NuminaMath or OpenMathInstruct into pre-training was not a priority for us in the current study. That said, we agree that adding instruction-tuned (SFT) data to pre-training could be a promising direction for further improving model performance. In fact, we believe that combining high-quality SFT datasets with pre-training data—such as incorporating OpenMathInstruct or NuminaMath—could potentially lead to even stronger results, and we consider this an exciting avenue for future work.

---

> ### Author Response · Authors · 2025-06-03
> **Response to Reviewer J7PA [1/3]**
>
> We sincerely thank you for recognizing MegaMath’s well-structured design studies, demonstrated effectiveness, and high reproducibility. We are also pleased to see your appreciation of MegaMath’s significant contributions to the open-source community. We are happy to address all of your questions and clarify any points that may have been misunderstood in your reviews.
>
> $\textrm{\color{blue}Question 1}$
> ```
> The paper does a thorough job of execution but lacks novelty.
> ```
> We appreciate the reviewer’s recognition of the thorough execution of our work. While our paper places significant emphasis on engineering and pipeline design, we would like to highlight several impactful contributions that go beyond implementation details.
>
> **First, MegaMath introduces one of the most comprehensive and large-scale math pretraining datasets to date—spanning over 370B tokens—alongside a highly scalable and fully reproducible pipeline. We believe this alone constitutes a valuable and timely resource, particularly given the ongoing limitations in academic access to high-quality pretraining data.**
>
> In terms of novelty, our focus lies not only in proposing novel techniques, but also in **developing effective, scalable, and impactful methods that lead to demonstrable improvements in both the scale and quality of training data**. Notable examples include:
>
> - **A two-stage text extraction strategy that achieves a practical balance between accuracy and efficiency**. Unlike prior approaches (e.g., OpenWebMath[1]; InfiMM-WebMath[2]) that typically rely on a single-pass extraction (which often forces a trade-off between scale and precision), our method decouples coarse and fine extraction steps, allowing us to efficiently process hundreds of billions of tokens while maintaining high data quality.
> - **Robust fasttext training**: MegaMath employs a robust fastText training pipeline by leveraging 2M high-quality examples labeled using Llama-3.1-70B and CoT data, coupled with an out-of-distribution evaluation suite spanning arXiv, Wikipedia, and textbook content. Through normalization and tuning, it achieves a 98.8% F1 score, ensuring reliable math-document classification at scale.
> - **Refinements to the data synthesis pipeline, along with novel filtering mechanisms**—such as AST-based filtering for code blocks to improve safety and correctness—which enhance the reliability and soundness of the dataset.
>
> We respectfully consider that the notion of novelty should be considered in light of practical scientific impact. In this sense, MegaMath provides not merely a dataset, but a comprehensive suite of data-centric curating pipelines designed to advance future research in math reasoning and pretraining data curating methodologies.
>
> [1] OpenWebMath: An Open Dataset of High-Quality Mathematical Web Text
>
> [2] InfiMM-WebMath-40B: Advancing Multimodal Pre-Training for Enhanced Mathematical Reasoning
>
> $\textrm{\color{blue}Question 2}$
> ```
> The impact of pretraining for math reasoning is not clear after post-training. Would the gains remain post-alignment as well?
> ```
>
> Thank you for your constructive suggestion. In response, we have conducted supervised fine-tuning (SFT) experiments on both the **Llama-3.2-3B-Base** and our **MegaMath-Llama3.2-3B-Base** using a random 5M subset of OpenMathInstruct2 (OMI2), as you recommended.
>
> To ensure a fair comparison, all models were trained with the same data, learning rate (i.e., 5e-6), number of epochs(i.e., 1), global batch size (i.e., 256), and other hyperparameters and then were evaluated in the same zero-shot manner.
> We present below a set of representative results across key math problem-solving benchmarks, including those reported in the MegaMath Core Eval set.
>
> As shown in the Table, the **MegaMath**-pretrained model consistently outperforms the baseline **Llama-3.2-3B-Base** after SFT, and also **Llama-3.2-3B-Instruct**, achieving stronger performance across multiple math benchmarks.
>
> We hope this helps clarify the value of pretraining on MegaMath and thus addresses your concern.
>
> | **Model**                 | **ASDIV** | **GSM8K** | **MATH500** | **MATH** | **MAWPS** | **OCW** | **SVAMP** | **TabMWP** |
> |----------------------------|-----------|------------|--------------|-----------|------------|----------|------------|------------|
> | Llama-3.2-3B-Instruct          | 78.1      | 67.5       | 32.6         | 32.8      | 90.4       | 7.7      | 81.4       | 37.6       |
> | Llama-3.2-3B-Base + SFT        | 85.8      | 73.1       | 41.4         | 45.6      | 93.1       | 7.0      | 79.8       | 65.9       |
> | **MegaMath-Llama-3.2-3B-Base + SFT** | **91.3** | **82.7**  | **52.6**    | **55.1** | **95.1**   | **13.6** | **86.2**  | **73.6**  |

---

> ### Comment · Area_Chair_Yvo5 · 2025-06-08
>
> Dear Reviewer J7PA,
>
> As we approach the end of the discussion period, could you please check the authors' responses and see if they have addressed your concerns? Thank you very much for your efforts.
>
> Best,
> AC

---

> ### Comment · Reviewer_J7PA · 2025-06-09
>
> Thanks for clarifying the definition of "information completeness".
>
> By CoT data, I indeed meant the first interpretation. The data has been mentioned multiple times in Section 2.1.3, figure 5, and Section 3.2 without any definition in the paper. I appreciate sharing the details, but this is an instance of bad writing to not even describe something which has been referenced at least 4 times in the main paper, and reference it by a generic name.
>
> > How do the different subsets of the dataset - web, code, and synthesized data?
> Bad writing on my part :P
> I meant to ask "how do the three subsets impact the performance?". It's a costly ablation, but just wanted to hear the authors' thoughts.
>
> I have increased the score by 1 given the discussions.

---

> > ### Author Response · Authors · 2025-06-10
> >
> > Thank you for your comments! We’re glad to see the previous issue is resolved, and we will definitely incorporate the missing details in the updated version.
> >
> > Regarding the follow-up question, we agree it’s an interesting direction:
> >
> > In Section 3.3 Ablation on MegaMath-Code, we also conducted some exploratory experiments along similar lines by mixing web and code data. Although both are math-related, they clearly play different roles in shaping model capabilities. For example, we find that code data more significantly improves the model’s ability to solve problems using code, whereas web data does not have the same effect but improve the CoT performance.
> >
> > Additionally, the impact of different data mixing ratios and the timing of mixing at various training stages on model performance remains an important factor. (For instance, high-quality synthetic data will be put in annealing phase of training with a higher ratio, such as MiniCPM, OLMo2)
> >
> > Overall, we believe that data mixture strategy is itself a valuable research direction. Several works (e.g., RegMix[1], CLIMB[2]) have begun modeling the data mixing problem in a principled and learnable way, such as formulating it as a regression task.
> > Through MegaMath, we also aim to provide a large-scale pool of mathematical data that can serve as a foundation for future data-centric research efforts.
> >
> > We’re glad to keep clarifying things to make sure all your concerns are resolved.
> >
> > [1] Regmix: Data mixture as regression for language model pre-training
> >
> > [2] CLIMB: CLustering-based Iterative Data Mixture Bootstrapping for Language Model Pre-training

---

### Author Response · Authors · 2025-06-03
**Overall Response**

We thank all reviewers for their constructive feedback. We’re encouraged by the shared recognition of our contributions, including the construction of an industrial-scale (371B) math pretraining corpus with broad utility (`R-J7PA`, `R-CT6F`, `R-PQwQ`), a well-executed and systematic data pipeline (`R-CT6F`, `R-PQwQ`), comprehensive ablation studies validating key design choices (`R-J7PA`, `R-CT6F`), strong empirical improvements on baseline models (`R-CT6F`), and the reproducibility and open-source value of our methods and data (`R-J7PA`). We have answered all reviewers' questions and updated more results accordingly.

**Summary of our updates:**
- Post-training experiments to demonstrate MegaMath's long-last effectiveness even on post-training phase (`R-J7PA`)
- The 100B-token training mixture configuration to train MegaMath-Llama and our selection principles (`R-CT6F`)
- Clarified that the CoT "discrepancy" originated from the conflation of base vs. instruct LLaMA models (`R-CT6F`)
- A consolidated LLM usage table across all annotation/synthesis stages (`R-PQwQ`)
- Supplemented MegaMath-Synthetic pre-training and evaluation results using the final version (Table 6 & 7) (`R-PQwQ`)

We hope these clarifications and additional experiments address the remaining concerns and further demonstrate the novelty, transparency, and significance of MegaMath.

---

### Decision · Program_Chairs · 2025-07-08

**Decision:**

Accept

**Comment:**

Summary:

This paper introduces MegaMath, a large-scale (371B tokens) mathematical pre-training corpus curated from web data, math-related code, and synthetic QA/code blocks. It provides a detailed construction pipeline and validates the dataset through ablation studies and empirical gains on multiple math reasoning benchmarks using LLaMA-3.2 models. The dataset and pipeline are intended as a public resource to enhance LLMs’ math reasoning capabilities.

Strengths:

All reviewers agree that the paper presents a valuable open resource. Most reviewers note the paper presents a well-documented engineering pipeline, and conducts comprehensive ablations and shows consistent performance gains on math benchmarks.

Weaknesses:

Originally, reviewers had concerns regarding limited novelty, limited evaluation using models beyond LLaMA-3.2 (unclear generalization to other strong models such as QWen), and limited evaluation on hard math benchmarks.

After the response period, the authors have addressed most concerns. The authors should make sure to include the discussions and new results in the revised paper. I do think that the impact of this paper could be enhanced if the proposed dataset can be used to improve the performance of more models beyond LLaMA-3.2. Nonetheless, I think the paper makes a valuable contribution and would recommend accepting it to the conference, provided that the paper could fully open-source all the resources as promised in the paper and during the author response period. In addition, I believe the authors should talk about the license issue when releasing this dataset. In particular, Stack-v2 has their user agreement terms that the paper may need to follow.